# An opioid-gated thalamoaccumbal circuit for the suppression of reward seeking in mice

Suppression of dangerous or inappropriate reward-motivated behaviors is critical for survival, whereas therapeutic or recreational opioid use can unleash detrimental behavioral actions and addiction. Nevertheless, the neuronal systems that suppress maladaptive motivated behaviors remain unclear, and whether opioids disengage those systems is unknown. In a mouse model using two-photon calcium imaging in vivo, we identify paraventricular thalamostriatal neuronal ensembles that are inhibited upon sucrose self-administration and seeking, yet these neurons are tonically active when behavior is suppressed by a fear-provoking predator odor, a pharmacological stressor, or inhibitory learning. Electrophysiological, optogenetic, and chemogenetic experiments reveal that thalamostriatal neurons innervate accumbal parvalbumin interneurons through synapses enriched with calcium permeable AMPA receptors, and activity within this circuit is necessary and sufficient for the suppression of sucrose seeking regardless of the behavioral suppressor administered. Furthermore, systemic or intra-accumbal opioid injections rapidly dysregulate thalamostriatal ensemble dynamics, weaken thalamostriatal synaptic innervation of downstream neurons, and unleash reward-seeking behaviors in a manner that is reversed by genetic deletion of thalamic μ-opioid receptors. Overall, our findings reveal a thalamostriatal to parvalbumin interneuron circuit that is both required for the suppression of reward seeking and rapidly disengaged by opioids.

Suppression of reward-seeking behaviors is critical for survival[1–3], whereas prescription or recreational opioid use can lead to unrestrained behavioral actions and opioid use disorder (OUD)[4]. Despite this knowledge, the neuronal systems that govern the suppression of reward-motivated behaviors remain unclear, and whether those systems are disengaged by opioids to unleash inappropriate behavioral actions is unknown. The paraventricular nucleus of the thalamus (PVT) is a candidate brain region underlying opioid-gated suppression of reward seeking, as it provides an interface for motivational circuits[5–7] and has among the highest level of μ-opioid receptors of any thalamic nucleus[8]. Recent studies demonstrate that stimulation of PVT neurons can blunt feeding behaviors and promote avoidance[9–14]. Furthermore, inhibition of PVT neurons that project to the nucleus accumbens (PVT→NAc) can invigorate food seeking when food is expected but

omitted[12,15]. Despite this knowledge, whether PVT→NAc circuitry provides a keystone neuronal substrate for the suppression of reward-motivated behavior, which could be provoked by a fear, stress, or inhibitory learning[16–20], is unknown. Furthermore, the neuronal mechanisms whereby opioids unleash maladaptive behavioral actions have not been identified.

Using deep-brain two-photon calcium imaging in vivo, we discover PVT→NAc neuronal ensembles that display activity patterns predictive of the expression and suppression of sucrose self-administration and seeking. Electrophysiological, optogenetic, and chemogenetic studies reveal that PVT→NAc neurons pervasively govern the suppression of sucrose self-administration and seeking through innervation of NAc parvalbumin interneurons and at synapses enriched with calcium-permeable AMPA receptors. Furthermore, we find that systemic or

✉e-mail: otis@musc.edu

intra-NAc opioid exposure attenuates PVT→NAc neuronal ensemble dynamics, weakens PVT→NAc synaptic innervation of downstream neurons, and rapidly unleashes sucrose seeking in a manner that is reversed by genetic deletion of thalamic μ-opioid receptors. Overall, our findings reveal a thalamostriatal feedforward inhibitory brake for reward seeking that can be rapidly disengaged by opioids.

## Results

### PVT→NAc ensemble dynamics track the expression and suppression of sucrose seeking

Here we designed a paradigm that allows two-photon calcium imaging in deep-brain regions during the expression and suppression of sucrose reward self-administration and seeking. Head-fixed mice were trained to press an active, but not an inactive, lever resulting in the presentation of a tone cue followed by a liquid sucrose reward

(Fig. 1a–c). Suppression of active lever pressing for sucrose could be incited by the fear-provoking predator odor TMT (2,5-dihydro-2,4,5-trimethylthiazoline)[21], by the pharmacological stressor yohimbine[22], or by extinction learning wherein the cue and sucrose were omitted (Fig. 1d–f and Supplementary Fig. 1)[20]. Reward seeking could also be reinstated through cue re-exposure after extinction training (Fig. 1g). To monitor activity in PVT→NAc neurons throughout this paradigm, mice received injections of a retrogradely trafficked virus encoding Cre-recombinase bilaterally into the NAc shell (rgAAV-hSyn-Cre) in combination with a Cre-inducible virus encoding a calcium indicator into the posterior PVT (AAVdj-DIO-GCaMP6m; Fig. 1h)[23]. Next, a microendoscopic GRIN lens was implanted dorsal to PVT, allowing chronic visual access to fluorescent GCaMP6m-expressing PVT→NAc projection neurons (Fig. 1i). Using two-photon imaging, we measured PVT→NAc neuronal activity dynamics around each active lever press

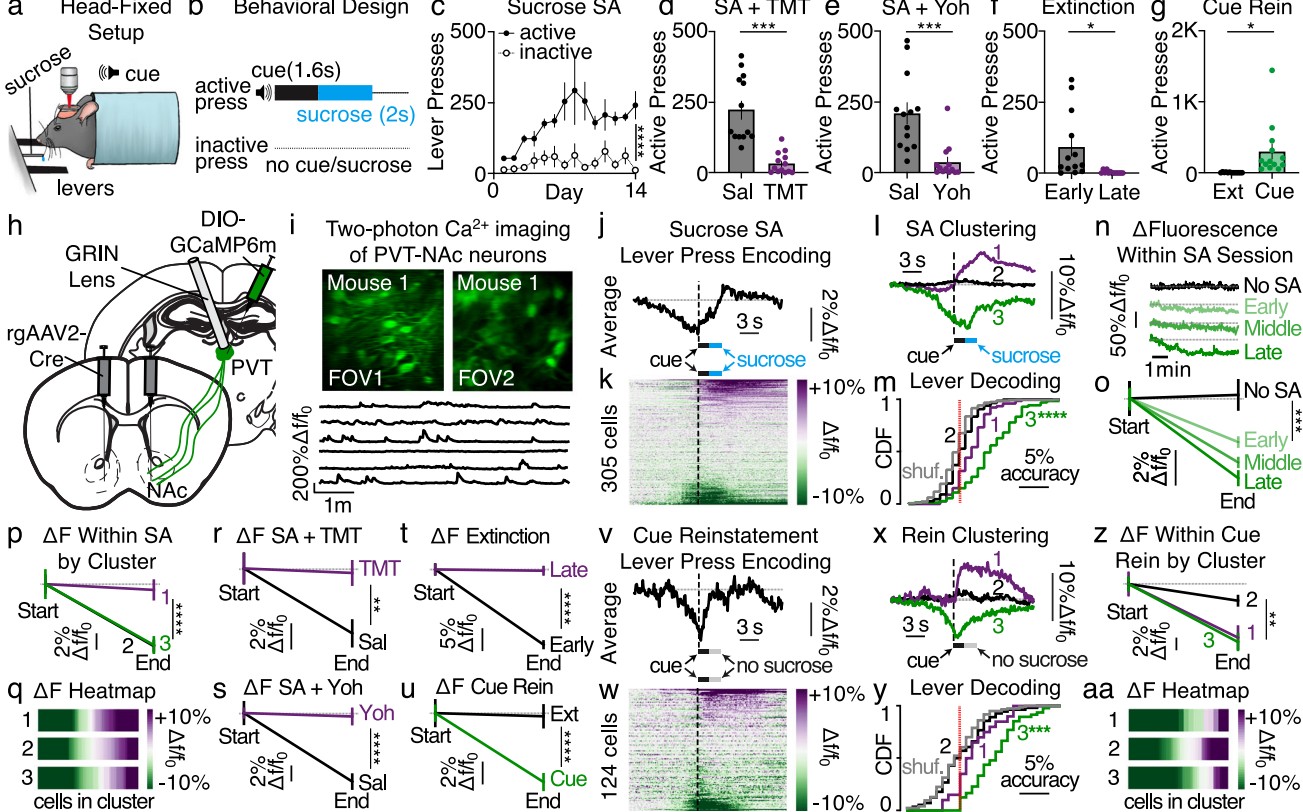

**Fig. 1 | Inhibition of select PVT→NAc neuronal ensembles predicts sucrose self-administration and seeking. a–c** Behavioral design (**a**; image modified from Vollmer et al., 2021[71]), schematic (**b**), and grouped data (**c**) for sucrose self-administration ($n = 13$ mice; two-way ANOVA, lever: $F_{1,24} = 60.65$, $P < 0.001$). **d–f** TMT (**d**), yohimbine (**e**), and extinction (**f**) suppressed lever pressing ($n = 13$ mice; TMT: two-tailed $t$-test $t_{12} = 5.54$, $P = 0.001$; yohimbine: $t_{12} = 5.66$, $P = 0.001$; extinction: $t_{12} = 2.72$, $P = 0.02$). **g** Cue presentation provoked reinstatement after extinction ($n = 13$ mice; $t_{12} = 2.79$, $P = 0.02$). **h, i** Surgery (**h**) for visualization of PVT→NAc neurons (**i**; top) and calcium-mediated fluorescent signals (**i**; bottom). **j, k** Averaged trace (**j**) and single-cell heatmap (**k**) revealing PVT→NAc dynamics during sucrose self-administration ($n = 6$ mice, 305 cells). **l** Clustering revealed three PVT→NAc neuronal ensembles induced by sucrose self-administration: excitatory (purple, 79 cells), non-responding (black, 153 cells), and inhibitory (green, 73 cells). **m** Active lever press decoding was most accurate for ensemble 3 during self-administration (two-way ANOVA, ensemble × shuffle interaction: $F_{2,604} = 34.02$, $P < 0.001$; Sidak's post-hoc: $P$s $< 0.001$). **n, o** Example waveforms (**n**) and grouped data (**o**) showing reduced PVT→NAc activity during self-administration but not a baseline (No-SA) session ($n = 3$–6 mice, 105–327 neurons; two-way ANOVA, session × time: $F_{3,2116} = 5.19$, $P = 0.001$; Sidak's post-hoc: No-SA vs. other sessions, $P$-values $< 0.002$). **p, q** Grouped data (**p**) and heatmap (**q**) reveal within-session

reductions in activity for each ensemble (two-way ANOVA, ensemble × time: $F_{2,604} = 4.99$, $P = 0.007$; Sidak's post-hoc: excited vs. inhibited/non-responding, $P$-values $< 0.001$). **r–t** TMT (**r**), yohimbine (**s**), and extinction (**t**) prevented reductions in PVT→NAc activity (66–150 neurons/session; TMT: two-way ANOVA, session × time: $F_{1,550} = 4.19$, $P = 0.04$, Sidak's post-hoc: $P = 0.002$; yohimbine: ANOVA, $F_{1,274} = 12.59$, $P < 0.001$, Sidak's post-hoc: $P < 0.001$; extinction: ANOVA, session × time: $F_{1,570} = 47.04$, $P < 0.001$, Sidak's post-hoc: $P < 0.001$). **u** PVT→NAc activity was reduced during cue-induced reinstatement ($n = 124$ cells; two-way ANOVA, session × time: $F_{1,544} = 7.83$, $P = 0.005$; Sidak's post-hoc: $P < 0.001$). **v, w** Averaged trace (**v**) and heatmap (**w**) revealing PVT→NAc dynamics during cue reinstatement. **x** Clustering revealed three ensembles: excitatory (20 cells), non-responding (56 cells), and inhibitory (48 cells). **y** Active lever press decoding for ensemble 3 during reinstatement was most accurate (two-way ANOVA, ensemble × shuffle: $F_{2,242} = 8.23$, $P = 0.0004$; Sidak's post-hoc: $P < 0.001$). **z, aa** Grouped data (**z**) and heatmap (**aa**) reveal reductions in activity for each ensemble during cue-induced reinstatement (two-way ANOVA, ensemble × time: $F_{2,242} = 4.27$, $P = 0.015$; Sidak's post-hoc: non-responding vs. excited/inhibited, $P$-values $< 0.004$). FOV field of view, SA self-administration, Rein reinstatement, Yoh yohimbine. Group comparisons: *$P < 0.05$, **$P < 0.01$, ***$P = 0.001$, ****$P < 0.001$. Bar and line graphs presented as mean ± SEM. Source data are provided as a Source Data file.

after sucrose self-administration behavior was established (late in learning, days 13–14). Data revealed that PVT→NAc neuronal activity was reduced at the population level upon active lever responding for sucrose (Fig. 1j), although cell-to-cell heterogeneity was apparent (Fig. 1k). Spectral clustering[24,25] identified three distinct PVT→NAc neuronal ensembles that could account for the response heterogeneity: those that were activated (ensemble 1), non-responding (ensemble 2), and inhibited (ensemble 3; Fig. 1l, and Supplementary Fig. 1). Machine learning based behavioral decoding revealed that the PVT→NAc population dynamics could be used to predict active lever pressing during each behavioral session (Supplementary Fig. 2). Additionally, while the activity of each neuronal ensemble could predict active lever pressing, the inhibitory dynamics of ensemble 3 provided superior decoding as compared with other ensembles (Fig. 1m). By tracking a subset of neurons across acquisition of sucrose self-administration, we determined that these ensemble-specific activity patterns develop across learning, as ensembles 1 and 3 displayed significant response adaptations from early to late self-administration training sessions (Supplementary Fig. 2). Next, we measured the change in basal PVT→NAc GCaMP6m fluorescence across each sucrose self-administration session, as basal fluorescence serves as a proxy for firing rates in tonically active cell populations[25,26] including PVT→NAc neurons[6]. Data revealed that within each sucrose self-administration session, the fluorescence of PVT→NAc neurons decreased across time (Fig. 1n, o). The within-session reduction was most pronounced after the behavioral task was well established (during late sessions; days 13–14) but was also present during earlier acquisition sessions (early sessions: days 1–2; middle sessions: days 7–8). Furthermore, the attenuation in activity was specific to ensembles 2 and 3 (Fig. 1p, q). Overall, both acute and tonic inhibition of select PVT→NAc neuronal ensembles predicts sucrose self-administration, consistent with the idea that activity in PVT→NAc neurons serves as a tonic 'brake' that must be released to initiate reward-motivated behavior.

We next monitored PVT→NAc neurons during the presentation of stimuli that suppress sucrose self-administration. The fear-provoking predator odor TMT and pharmacological stressor yohimbine not only prevented active lever pressing, but also prevented the inhibition of PVT→NAc neurons during sucrose self-administration (Fig. 1r, s). Additionally, while PVT→NAc neurons were inhibited when sucrose was omitted during early extinction sessions (days 1–2), this inhibition was prevented during late sessions after extinction learning was established (days 9–10; Fig. 1t). Thus, external stimuli and inhibitory learning that suppress sucrose self-administration and seeking also prevent the tonic inhibition of PVT→NAc neurons. Finally, we determined if PVT→NAc neuronal inhibition could be reinstated after extinction during a cue-induced reinstatement test. Indeed, PVT→NAc neurons became tonically inhibited during cue-induced reinstatement (Fig. 1u) and showed acute, heterogeneous responses upon active lever pressing despite the absence of sucrose (Fig. 1v–w). Spectral clustering revealed qualitatively similar neuronal ensembles during cue-induced reinstatement (Fig. 1x, and Supplementary Fig. 1) as compared with previous sucrose self-administration sessions, with the inhibitory dynamics of ensemble 3 providing superior decoding of active lever pressing as compared with other ensembles (Fig. 1y–aa). Overall, sucrose self-administration and seeking are associated with ensemble-specific inhibition of PVT→NAc neurons, whereas competing behavioral suppressors prevent that inhibition. Despite these findings, whether activity in PVT→NAc is both necessary and sufficient for the suppression of reward seeking, regardless of behavioral suppressor, remains unknown.

## PVT→NAc neuronal activity is necessary and sufficient for the suppression of sucrose seeking

We next examined the function of PVT→NAc neuronal activity for the expression and suppression of sucrose self-administration and seeking. PVT→NAc neurons were targeted for optogenetic manipulation through injections of a retrogradely trafficked virus encoding Cre-recombinase bilaterally into the NAc shell (rgAAV2-hSyn-Cre) in combination with viruses encoding Cre-inducible channelrhodopsin (AAV5-DIO-Ef1α-ChR2), halorhodopsin (AAV5-DIO-Ef1α-eNpHR3.0), or control enhanced yellow fluorescence protein (AAV5-DIO-Ef1α-eYFP) into the posterior PVT (Fig. 2a, b). During sucrose self-administration (Fig. 2c), optogenetic stimulation of PVT→NAc neurons in ChR2 mice decreased active lever pressing (Fig. 2d, e) akin to previous findings[11,15]. Furthermore, while the predator odor TMT or pharmacological stressor yohimbine suppressed sucrose self-administration in control eYFP mice, inhibition of PVT→NAc neurons prevented that behavioral suppression in eNpHR mice (Fig. 2f, g). Similarly, while extinction training suppressed active lever pressing in the absence of sucrose, the inhibition of PVT→NAc neurons unleashed active lever pressing (Fig. 2h). Finally, after extinction learning we found that cue-induced reinstatement of sucrose seeking was suppressed by the stimulation of PVT→NAc neurons (Fig. 2i). Importantly, the behavioral effects of PVT→NAc stimulation and inhibition were specific to the active but not inactive lever (Supplementary Fig. 3), confirming that the optogenetic manipulations affected behavior in a lever-specific manner. In further support of this specificity, we found that PVT→NAc stimulation did not inhibit locomotor activity, despite provoking a real-time place aversion (Supplementary Fig. 4) as previously reported[13]. Together, these data reveal that PVT→NAc neuronal activity is both necessary and sufficient for the suppression of sucrose self-administration and seeking, an effect that is generalizable across behavioral suppressors.

## PVT→NAc-dependent suppression of sucrose seeking requires downstream CP-AMPARs and PV interneurons

We next identified candidate downstream cellular targets that may underlie thalamostriatal-dependent suppression of reward seeking. Following injections of an anterogradely trafficked transsynaptic AAV1 virus expressing Cre (AAV1-CamKII-Cre)[27–29] in the posterior PVT, we infused a Cre-dependent virus encoding eYFP bilaterally in the NAc (AAV5-ef1α-DIO-eYFP). Following incubation, we performed immunohistochemistry for eYFP in combination with immunohistochemistry for known NAc cell types in ex vivo brain slices (Supplementary Fig. 5). Overall, experiments confirmed that putative dopamine 1 receptor and dopamine 2 receptor expressing medium spiny neurons in the NAc shell are synaptically innervated by PVT (PVT→NAc[D1-MSNs]; PVT→NAc[D2-MSNs]). We also found elevated anterograde transsynaptic labeling of parvalbumin-expressing interneurons in the NAc shell (PVT→NAc[PV-INs]) as compared with other striatal interneurons, a finding consistent with a previous electrophysiological study showing functional input from PVT to striatal PV interneurons[30]. To further characterize PVT synaptic innervation of D1-MSNs, D2-MSNs, and PV-INs, we injected a virus encoding the red-shifted excitatory opsin ChrimsonR (AAV5-hSyn-ChR)[31] into PVT and a Cre-inducible virus encoding eYFP (AAV5-ef1α-DIO-eYFP) into the NAc shell of Cre-driver mouse lines (D2-Cre; PV-Cre; Fig. 3a, b). Subsequent patch-clamp electrophysiological recordings revealed elevated glutamatergic excitatory synaptic drive at PVT→NAc[PV-IN] synapses as compared with PVT→NAc[D1-MSN] and PVT→NAc[D2-MSN] synapses (Fig. 3c, d). Additionally, we found that PVT→NAc[PV-IN] synapses were inwardly rectifying, whereas other synapses were not (Fig. 3e, f). Inwardly rectifying glutamatergic synapses suggest the presence of calcium-permeable AMPA receptors (CP-AMPARs), which are known to be selectively enriched in striatal PV interneurons[32]. Indeed, we found that pharmacological inhibition of CP-AMPARs selectively attenuated excitatory synaptic drive at PVT→NAc[PV-IN] synapses, but not PVT→NAc[D2-MSN] or putative PVT→NAc[D1-MSN] synapses (Fig. 3g, h). Altogether, PVT→NAc projection neurons have biased innervation of PV interneurons at synapses enriched with CP-AMPARs, however the function of this synaptic connectivity for the suppression of reward seeking remains unknown.

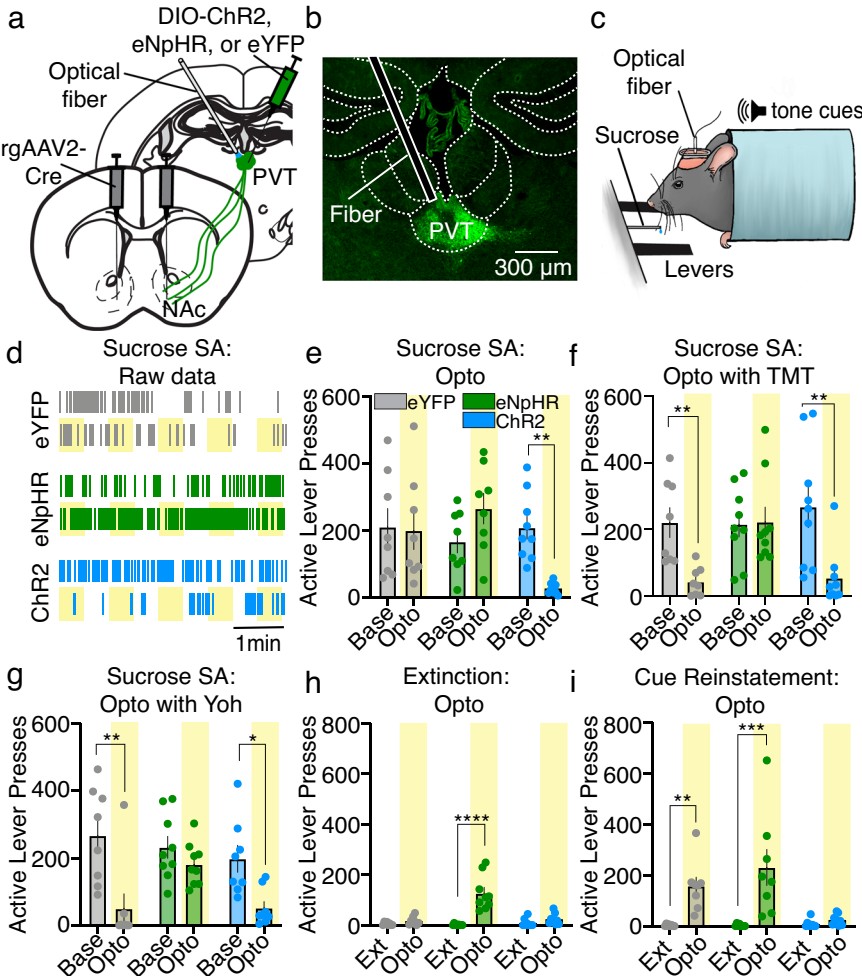

**Fig. 2 | PVT→NAc activity dynamics are necessary and sufficient for the expression and suppression of sucrose self-administration and seeking.** **a–c** Surgical strategy (**a**) for optogenetic manipulation of PVT→NAc neurons (**b**) during sucrose self-administration (**c**; image modified from Vollmer et al.[71]). **d** Raster plot showing example active lever pressing rates in each group during sucrose self-administration (examples from 5 mice/group; yellow bar = light on). **e** Group data showing that optogenetic stimulation of PVT→NAc neurons suppressed active lever pressing ($n = 8$ eYFP, 8 eNpHR, 9 ChR2 mice; repeated-measures two-way ANOVA, day × group interaction: $F_{2,22} = 7.09$, $P = 0.004$; Sidak's post-hoc: $P = 0.006$). **f–h** TMT (**f**), yohimbine (**g**), and extinction (**h**) suppressed active lever pressing, whereas inhibition of PVT→NAc neurons in eNpHR mice rescued active lever pressing (TMT: $n = 8$ eYFP, 9 eNpHR, 9 ChR2 mice; repeated-measures two-way ANOVA, day × group

interaction: $F_{2,23} = 5.36$, $P = 0.01$; Sidak's post-hoc: eYFP $P = 0.009$, ChR2 $P = 0.001$; yohimbine: $n = 8$ eYFP, 9 eNpHR, 8 ChR2 mice; repeated-measures two-way ANOVA, day effect: $F_{1,22} = 20.46$, $P = 0.002$; Sidak's post-hoc: eYFP $P = 0.002$, ChR2 $P = 0.04$; extinction: $n = 8$ eYFP, 9 eNpHR, 9 ChR2 mice; repeated-measures two-way ANOVA, day × group interaction: $F_{2,23} = 19.55$, $P < 0.001$; Sidak's post-hoc: $P < 0.001$). **i** Cue-induced reinstatement of active lever pressing after extinction was abolished by stimulation of PVT→NAc neurons in ChR2 mice ($n = 8$ eYFP, 8 eNpHR, 9 ChR2 mice; repeated-measures two-way ANOVA, day × group interaction: $F_{2,22} = 6.15$, $P = 0.008$; Sidak's post-hoc: eYFP $P = 0.007$, eNpHR $P = 0.001$). Ext extinction, Opto optogenetic manipulation, SA self-administration. Group comparisons: *$P < 0.05$, **$P < 0.01$, ***$P = 0.001$, ****$P < 0.001$. Bar graphs are presented as mean ± SEM. Source data are provided as a Source Data file.

We next determined the necessity of cell-type biased signaling mechanisms for PVT→NAc-dependent suppression of reward seeking. To do so, we replicated the above behavioral optogenetics experiment (shown in Fig. 2) but implanted a bilateral cannula dorsal to the NAc shell allowing simultaneous neuropharmacological manipulation (Fig. 3i). Optogenetic stimulation of PVT→NAc neurons suppressed sucrose self-administration as above, and pharmacological inhibition of D1 or D2-receptor signaling did not rescue lever pressing behavior. In contrast, inhibition of NAc CP-AMPARs selectively rescued active lever pressing during optogenetic stimulation of PVT→NAc neurons (Fig. 3j, and Supplementary Fig. 6). Considering that PVT→NAc^PV-IN synapses are enriched with CP-AMPARs, we next used chemogenetics to determine the necessity of PV-interneuron activation for the suppression of sucrose self-administration (Fig. 3k). Following a similar surgical design as above, we injected a Cre-inducible virus encoding an inhibitory DREADD (AAV5-hSyn-DIO-hM4D(Gi)-mCherry) bilaterally in

the NAc of transgenic PV-Cre mice, a strategy that allows selective inhibition of accumbal PV interneurons (see Supplementary Fig. 6). In sucrose-trained mice, chemogenetic inhibition of NAc PV interneurons prevented the suppression of self-administration caused by optogenetic stimulation of PVT→NAc neurons (Fig. 3l), by presentation of the predator odor TMT (Fig. 3m), by the pharmacological stressor yohimbine (Fig. 3n), and by extinction learning (Fig. 3o). In contrast, these manipulations did not affect inactive lever pressing (Supplementary Fig. 6). Thus, PVT→NAc-dependent suppression of reward seeking requires activity of downstream CP-AMPARs and PV interneurons. While we therefore hypothesize that PVT→NAc neurons act selectively at CP-AMPAr-enriched synapses on PV interneurons to suppress behavior, it should be noted that other synapses and NAc cell types may also be involved (see discussion). Whether this feedforward inhibitory circuit for the suppression of reward seeking is modulated by opioids, however, remains unclear.

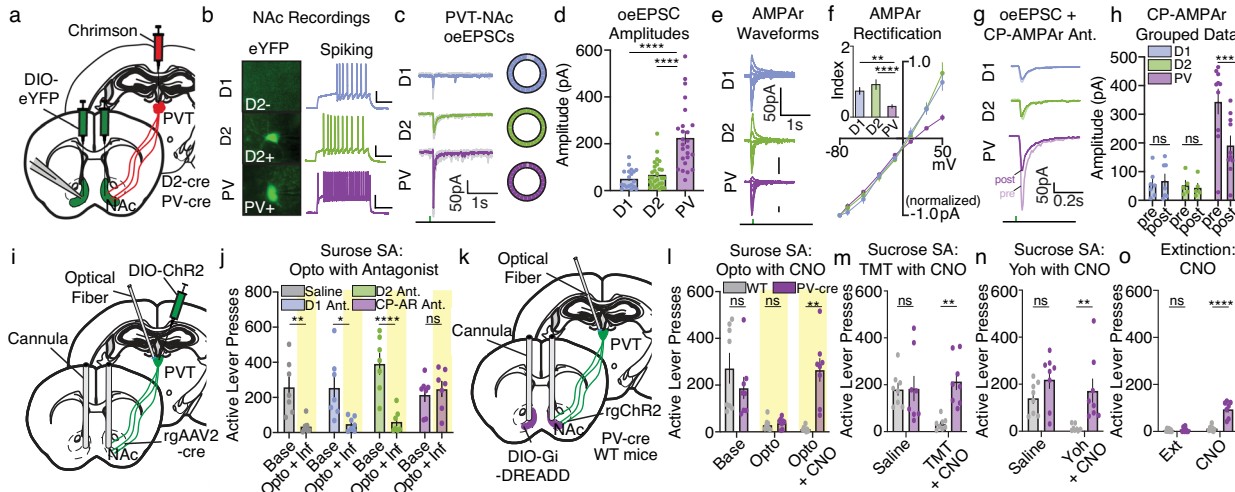

**Fig. 3 | PVT→NAc-dependent suppression of sucrose seeking requires down-stream CP-AMPARs and PV interneurons. a** Surgery for slice electrophysiology. **b** Example fluorescent images (left) and current-clamp traces (right) for NAc cell types (scale: 25 mV/0.2 s). **c, d** Example waveforms (**c**; pie charts show proportion of light-responding neurons) and grouped data (**d**) reveal elevated oeEPSC amplitudes in NAc PV-INs (D1-MSNs: $n = 19$ cells, 8 mice; D2-MSNs: $n = 28$ cells, 11 mice; PV-INs: $n = 25$ cells, 9 mice; one-way ANOVA, $F_{2,69} = 27.78$, $P < 0.001$; Sidak's post-hoc: PV-INs vs D1/D2-MSNs $P$-values < 0.001). **e, f** Example waveforms (**e**) and grouped data (**f**) showing PVT→NAc$^{PV-IN}$ synapses are selectively inwardly rectifying (D1-MSNs: $n = 11$ cells, 5 mice; D2-MSNs: $n = 10$ cells, 5 mice; PV-INs: $n = 16$ cells, 7 mice). Inset: rectification index ($I_{50}/I_{-70}$; one-way ANOVA, $F_{2,34} = 13.27$, $P < 0.001$; Sidak's post-hoc: PV-INs vs D1/D2-MSNs $P$-values < 0.003). **g, h** Waveforms (**g**) and grouped data (**h**) showing that bath application of the CP-AMPAr antagonist IEM-1640 selectively reduced oeEPSC amplitudes at PVT→NAc$^{PV-IN}$ synapses (D1-MSNs: $n = 7$ cells, 5 mice; D2-MSNs: $n = 5$ cells, 4 mice; PV-INs: $n = 10$ cells, 5 mice; two-way ANOVA, cell type × time: $F_{2,19} = 13.17$, $P = 0.003$; Sidak's post-hoc: PV-INs $P < 0.001$). **i** Surgery for simultaneous optogenetic manipulation of PVT→NAc neurons and intra-NAc neuropharmacology. **j** Microinfusions of the CP-AMPAr antagonist

prevented the suppression of sucrose self-administration caused by stimulation of PVT→NAc neurons ($n = 7$ mice/group; two-way ANOVA, group × day: $F_{3,24} = 5.98$, $P = 0.003$; Sidak's post-hoc: $P$-values < 0.01). **k** Surgery for simultaneous optogenetic stimulation of PVT→NAc neurons and chemogenetic inhibition of PV-INs. **l** CNO-mediated inhibition of NAc PV-INs prevented the suppression of sucrose self-administration caused by optogenetic stimulation of PVT→NAc neurons ($n = 8$ mice/group; two-way ANOVA, group x day: $F_{2,28} = 11.33$, $P < 0.001$; Sidak's post-hoc: Opto + CNO $P = 0.004$). **m–o** Chemogenetic inhibition of PV-INs also prevented the suppression of sucrose self-administration caused by TMT (**m**), yohimbine (**n**), and extinction learning (**o**) (TMT: $n = 8$ mice/group; two-way ANOVA, group × day: $F_{1,14} = 5.34$, $P = 0.04$; Sidak's post-hoc: TMT + CNO $P = 0.002$; yohimbine: $n = 8$ mice/group; ANOVA, group: $F_{1,14} = 10.93$, $P = 0.01$; Sidak's post-hoc: yohimbine + CNO $P = 0.005$; extinction: $n = 7$ mice/group; ANOVA, group × day: $F_{1,12} = 29.91$, $P = 0.001$; Sidak's post-hoc: extinction + CNO $P < 0.001$). CP-AMPAr calcium-permeable AMPA receptor, oeEPSC optically evoked excitatory postsynaptic current, SA self-administration, WT wild-type, Yoh yohimbine; Group comparisons: *$P < 0.05$, **$P < 0.01$, ****$P < 0.001$. Bar and line graphs are presented as mean ± SEM. Source data are provided as a Source Data file.

## PVT→NAc-dependent suppression of reward seeking is gated by μ-opioid receptors

Opioid use disorder is associated with maladaptive reward-seeking behaviors despite negative consequences[33–36]. Despite this knowledge, whether opioids can disrupt PVT→NAc-dependent behavioral inhibition to unleash reward seeking in the face of competing stimuli remains unknown. To investigate this possibility, we monitored PVT→NAc activity dynamics using two-photon calcium imaging (Fig. 4a) after mice were given a single systemic injection of the opioid heroin. Heroin reliably reduced the activity of these neurons (Fig. 4b, c). Furthermore, when heroin was administered prior to a sucrose self-administration session, PVT→NAc neuronal ensembles were altered such that a smaller proportion of neurons displayed excitatory (ensemble 1) or inhibitory (ensemble 3) dynamics (Fig. 4d, e, and Supplementary Fig. 7). To further investigate how PVT→NAc response dynamics were modulated by heroin, we tracked the activity of a subset of PVT→NAc neurons across sessions (Fig. 4f). Clustering of the tracked cells revealed that single-cell ensemble dynamics were altered (Fig. 4g), such that activated and inhibited clusters had a significant response attenuation upon heroin injection (Fig. 4h). Further analysis of tracked neurons revealed poorer active lever press decoding at both the population level (Fig. 4i) and at the level of individual neuronal ensembles (Fig. 4j). We next determined the influence of heroin on PVT→NAc-dependent behavioral suppression. Using the optogenetic viral strategy described above (Fig. 4k), we found heroin not only prevented the suppression of sucrose self-administration caused by optogenetic stimulation of PVT→NAc neurons (Fig. 4l), but also by the predator odor TMT (Fig. 4m) and pharmacological stressor yohimbine

(Fig. 4n). Heroin did not have a significant effect on inactive lever responding across conditions (Supplementary Fig. 7). Surprisingly, heroin also caused PVT→NAc stimulation to be appetitive rather than aversive, as mice expressed a stimulation-dependent real-time place preference (Supplementary Fig. 4). Together, these data suggest that systemic opioids may be modulating PVT→NAc neuronal activity, such that sucrose self-administration is promoted, rather than inhibited, in the presence of behavioral suppressors. However, the mechanism whereby opioids allow sucrose self-administration despite PVT→NAc activity remains unclear.

To determine how heroin may be modulating PVT→NAc neuronal activity and behavior, we first confirmed the presence of μ-opioid receptors (μ-ORs) on PVT→NAc neurons and validated a strategy to knockout these receptors. We selectively labeled PVT→NAc neurons by injecting a retrogradely trafficked virus encoding Cre-recombinase bilaterally into the NAc shell (rgAAV2-hSyn-Cre) and Cre-inducible eYFP into the posterior PVT (AAV5-DIO-hSyn-eYFP). Immunohistochemistry suggested the co-expression of μ-ORs receptors proximal to or overlapping with PVT→NAc somata. Furthermore, co-expression was reduced in transgenic $Oprm1^{fl/fl}$ mice which have loxP sites flanking exons 2–3 of the $Oprm1$ gene[37] allowing Cre-dependent knockout of PVT μ-ORs (Supplementary Fig. 8). Additionally, while we found that the selective μ-OR agonist DAMGO reduced the evoked firing rate of PVT neurons by over 50%, Cre-dependent knockout of PVT μ-ORs prevented DAMGO from reducing PVT firing rates (Supplementary Fig. 9).

We performed surgeries that would allow for optogenetic stimulation of PVT→NAc neurons along with Cre-dependent knockout of PVT

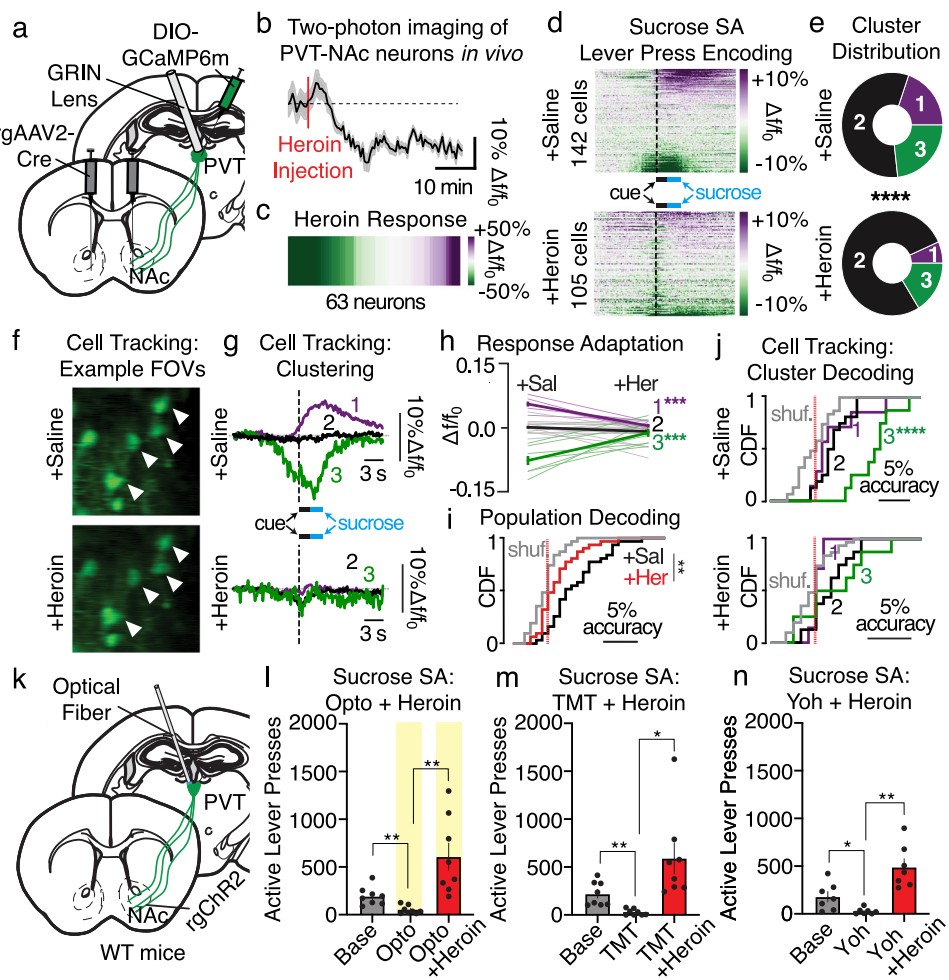

**Fig. 4 | PVT→NAc-dependent suppression of reward seeking is disrupted by an injection of heroin. a** Surgery for in vivo two-photon imaging. **b, c** Averaged trace (**b**) and heatmap (**c**) from two-photon imaging reveal that heroin reduced PVT→NAc neuronal activity ($n = 63$ neurons/3 mice; two-tailed t-test, $t_{62} = 3.03$, $P = 0.004$). **d** Heatmaps for all neurons during sucrose self-administration following injection of saline (top; $n = 142$ cells/4 mice) or heroin (bottom; $n = 105$ cells/4 mice). **e** Heroin reduced the proportion of cells in excited and inhibited ensembles (Chi-squared, $\chi^2 = 23.5$, $P < 0.001$). **f** Example FOVs for PVT→NAc neurons tracked across sucrose self-administration days ($n = 4$ mice, 31 tracked cells; top: saline, bottom: heroin). **g** Clustering of tracked cells revealed a change in ensemble dynamics for excitatory responders (7 cells), non-responders (16 cells), and inhibitory responders (8 cells). **h** Response amplitudes for tracked cells reveal significant response adaptations in ensemble 1 and 3, but not 2, due to heroin injection (two-way ANOVA, ensemble × time: $F_{2,56} = 29.21$; $P < 0.001$; Sidak's post-hoc: ensembles 1, 3 P-values = 0.001). **i** Heroin reduced active lever press decoding by PVT→NAc cells (two-way ANOVA, drug × shuffling: $F_{1,120} = 7.15$, $P = 0.01$; Sidak's post-hoc: Saline vs. Heroin $P = 0.002$). **j** Cluster decoding of tracked cells shows that the inhibitory ensemble best predicts an active lever during the saline test (top; two-way ANOVA, ensemble × shuffling interaction: $F_{2,56} = 7.00$, $P = 0.002$; Sidak's post-hoc: $P < 0.001$), but none of the ensembles can predict an active lever press during the heroin test (bottom; interaction: two-way ANOVA, $F_{2,56} = 0.75$, $P > 0.48$). **k** Surgical strategy for optogenetic manipulation. **l–n** Heroin prevented the suppression of sucrose self-administration caused by PVT→NAc stimulation (**l**), TMT (**m**), and yohimbine (**n**) (Opto: $n = 8$ mice/group; one-way ANOVA, $F_{2,21} = 11.56$, $P = 0.004$; planned two-tailed t-tests: base vs. opto $P < 0.01$, opto vs. heroin + opto $P < 0.01$; TMT: $n = 8$ mice/group; one-way ANOVA, $F_{2,21} = 8.77$, $P = 0.002$; planned two-tailed t-tests: base vs. TMT $P < 0.01$, TMT vs. heroin + TMT $P < 0.05$; yohimbine: $n = 7$ mice/group; one-way ANOVA, $F_{2,18} = 16.36$, $P < 0.001$; planned two-tailed t-tests: base vs. yohimbine $P < 0.05$, yohimbine vs. heroin + yohimbine $P < 0.01$). FOV field of view, SAL saline, HER heroin, Base Baseline, Opto optogenetics, Yoh Yohimbine. Group comparisons: *$P < 0.05$, **$P < 0.01$, ***$P = 0.001$, ****$P < 0.001$. Bar and line graphs are presented as mean values ± SEM. Source data are provided as a Source Data file.

μ-ORs (Fig. 5a). In situ hybridization via RNAscope confirmed a knockout of the μ-OR gene, but not kappa(κ)-OR gene, in the PVT of *Oprm1*[fl/fl] mice versus wild-type (WT) control mice (Fig. 5b, c). Next, these mice underwent sucrose self-administration training followed by a battery of behavioral suppression tests. As before, in WT mice we found that systemic heroin injection caused behavioral disinhibition despite optogenetic stimulation of PVT→NAc neurons (Fig. 5d), TMT exposure (Fig. 5e), or yohimbine injection (Fig. 5f). However, knockout of PVT μ-ORs in *Oprm1*[fl/fl] mice abolished heroin-induced behavioral disinhibition (Fig. 5d–f). Inactive lever pressing did not change across groups or conditions (Supplementary Fig. 10). Overall, we found that a single systemic injection of the opioid heroin prevents PVT→NAc-dependent suppression of sucrose seeking, an effect that requires PVT μ-ORs.

We next determined whether local NAc μ-OR activity can allow opioid-dependent behavioral disinhibition. To do so, we replicated previous behavioral optogenetics experiments (shown in Figs. 2 and 3) but implanted a bilateral cannula dorsal to the NAc shell, allowing for simultaneous pharmacological stimulation of μ-ORs using DAMGO (Fig. 6a). Optogenetic stimulation of PVT→NAc neurons suppressed sucrose self-administration as above, however, this effect was blocked by intra-NAc infusion of DAMGO (0.02 μg in 0.3 μL; Fig. 6b). Furthermore, DAMGO prevented the behavioral suppression caused by predator odor TMT exposure (Fig. 6c), and by the pharmacological stressor yohimbine (Fig. 6d). We next investigated whether PVT synaptic innervation of NAc MSNs (PVT→NAc[MSNs]) and PV interneurons was sensitive to local μ-OR activity. Similar to the viral strategy above

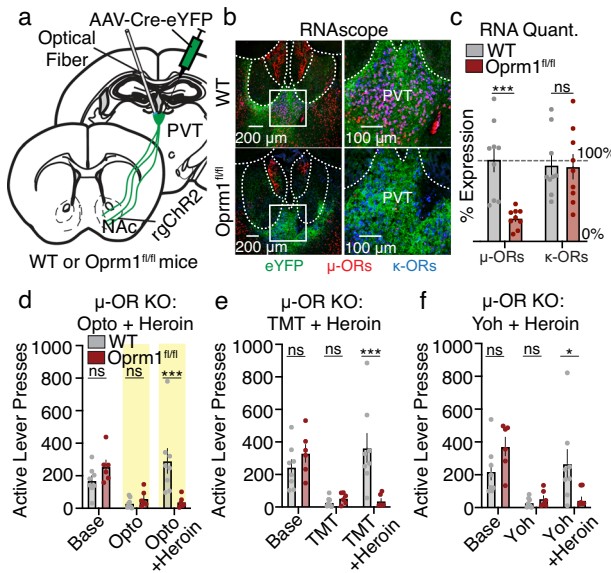

**Fig. 5 | Selective knockout of PVT μ-opioid receptors prevents heroin-induced behavioral disinhibition. a** Surgical strategy for PVT μ-OR knockout with simultaneous optogenetic manipulation of PVT→NAc neurons. **b** Example images showing RNAscope in situ hybridization of PVT μ-ORs and κ-ORs in WT (top) and *Oprm1fl/fl* (bottom) mice. **c** Quantification reveals reduced μ-OR but not κ-OR RNA expression in *Oprm1fl/fl* mice (n = 3 *Oprm1fl/fl*, 2 WT mice; two-way ANOVA, group × receptor interaction: $F_{1,32} = 7.61$, $P = 0.01$; Sidak's post-hoc: $P = 0.001$). **d–f** Knockout of PVT μ-ORs in *Oprm1fl/fl* mice rescued the suppression of sucrose self-administration caused by optogenetic stimulation of PVT→NAc neurons (**d**), TMT (**e**), and yohimbine (**f**) despite systemic heroin injection (n = 6–8 mice/group; Opto: n = 6 *Oprm1fl/fl*, 8 WT mice; repeated-measures two-way ANOVA, group × day interaction: $F_{2,24} = 8.33$, $P = 0.002$; Sidak's post-hoc: opto + heroin $P = 0.001$; TMT: n = 6 *Oprm1fl/fl*, 8 WT mice; repeated-measures two-way ANOVA, group × day interaction: $F_{2,24} = 8.94$, $P = 0.001$; Sidak's post-hoc: TMT + heroin $P = 0.001$; yohimbine: n = 6 *Oprm1fl/fl*, 8 WT mice; repeated-measures two-way ANOVA, group × day interaction: $F_{2,24} = 5.53$, $P = 0.01$; Sidak's post-hoc: TMT + heroin $P = 0.02$). μ-OR μ-opioid receptor, κ-OR κ-opioid receptor, Base Baseline, Opto optogenetics, Yoh Yohimbine. Group comparisons: *$P < 0.05$, **$P < 0.01$, ***$P = 0.001$. Bar graphs are presented as mean values ± SEM. Source data are provided as a Source Data file.

(see Fig. 3), we injected the excitatory opsin ChrimsonR (AAV5-hSyn-ChrR)[31] into PVT and a Cre-inducible virus encoding eYFP (AAV5-ef1α-DIO-eYFP) into the NAc shell of D2-Cre or PV-Cre transgenic mice (Fig. 6e). Using ex vivo patch-clamp electrophysiology, we found that bath application of DAMGO decreased glutamatergic excitatory synaptic drive at PVT→NAcMSN and PVT→NAcPV-IN synapses (Fig. 6f), suggesting that μ-OR activity may prevent PVT→NAc-dependent suppression of reward seeking through a presynaptic mechanism. To determine if knocking out PVT μ-ORs would rescue excitatory input onto accumbal MSNs and PV-INs, *Oprm1fl/fl* mice received a combinatorial injection of Cre-recombinase (AAV5-hSyn-Cre) and ChrimsonR (AAV5-hSyn-ChrR)[31] into PVT (1:10 ratio), and an injection of PV-interneuron targeted GCaMP6f (pAAV-S5E2-GCaMP6f)[38] bilaterally into NAc shell (Fig. 6g). Indeed, patch-clamp recordings confirmed that knockout of PVT μ-ORs prevented the DAMGO-induced attenuation of excitatory drive at putative PVT→NAcMSN synapses and PVT→NAcPV-IN synapses (Fig. 6h). Lastly, we investigated whether presynaptic μ-OR activity was necessary for driving opioid-induced behavioral disinhibition. A similar surgical strategy was used (see Fig. 5), wherein *Oprm1fl/fl* and WT mice received bilateral injections of a retrogradely trafficked channelrhodopsin (rgAAV2-hSyn-ChR2-eYFP) into NAc shell, and an injection of a Cre-recombinase (AAV5-hSyn-Cre) into PVT. Additionally, an optical fiber was implanted dorsal to PVT and a cannula was implanted dorsal to NAc (Fig. 6i). As before, an intra-NAc

infusion of DAMGO prevented the suppression of sucrose self-administration caused by optogenetic stimulation (Fig. 6j), TMT (Fig. 6k, and yohimbine (Fig. 6l) in WT mice. However, knockout of PVT μ-ORs in *Oprm1fl/fl* mice rescued PVT→NAc stimulation-, TMT-, and yohimbine-dependent behavioral suppression (Fig. 6j–l) despite NAc DAMGO infusion. Inactive lever pressing did not change across groups or conditions (Supplementary Fig. 10). Overall, these data reveal that PVT→NAc neurons and PVT→NAc-dependent behaviors may be tightly regulated by both somatic and presynaptic PVT→NAc μ-ORs.

## Discussion

Here we identify PVT→NAc neuronal ensembles that have acute and tonic activity patterns predictive of the expression and suppression of sucrose self-administration and seeking. These activity dynamics are necessary and sufficient for the suppression of sucrose seeking, causally mediated through downstream PV interneurons and CP-AMPArs which are enriched at PVT→NAcPV-IN synapses. We find that systemic or local NAc opioid injections reduce both PVT→NAc neuronal ensemble dynamics and synaptic innervation of downstream MSNs and PV-INs, unleashing sucrose seeking in the face of competing behavioral suppressors. However, knockout of PVT μ-ORs rescues PVT→NAc-mediated behavioral inhibition, despite systemic or intra-NAc opioid administration. Overall, we discover a keystone neuronal system for the suppression of reward-motivated behaviors and find that this system is rapidly disengaged by opioids.

Akin to our findings, previous studies show that stimulation of PVT can prevent feeding and promote avoidance[9–14], whereas inhibition of PVT neurons can promote reward seeking when food is expected but omitted[12,15]. Surprisingly, here we find that PVT→NAc neurons play a pervasive role in the suppression of reward-motivated behavior, and through an unexpected synaptic input to NAc PV interneurons. Other studies have focused on the role of PVT synaptic input to NAc D1- and D2-MSNs[13,39,40], and have found unique functions of these pathways in other behaviors such as the expression of opioid withdrawal[13] and the retrieval of an opioid conditioned place preference memory[40]. Thus, discrete outputs from PVT to unique NAc cell types likely control distinct behavioral states, including the suppression of reward seeking as described here.

Our electrophysiological data show that accumbal PV interneurons, as compared to putative D1- and D2-MSNs, receive elevated excitatory drive from PVT neurons, although there are potential caveats to our viral targeting techniques. First, we used D2-Cre and PV-Cre transgenic mice to target MSNs or PV interneurons, respectively, which could have led to variability in ChrimsonR expression between groups of animals. Second, in our D2-Cre transgenic mice, we classified non-fluorescent neurons as putative D1-MSNs, whereas these cells could have been unlabeled D2-MSNs or other cell populations. Despite these caveats, our findings are consistent with previous literature showing that accumbal fast-spiking interneurons (FSIs) receive greater excitatory input from PVT as compared with undefined MSNs using a within-subject design[30]. However, further studies comparing PVT synaptic input to each specific cell-type, including other subclasses of interneurons, within subjects could improve our understanding of PVT→NAc circuit biology.

Our data support the idea that accumbal PV interneurons and CP-AMPArs are necessary for the suppression of sucrose self-administration. Previously, others have shown that accumbal PV interneurons, as well as other FSIs within NAc, can act as powerful regulators of local neuronal activity and behavior despite being sparsely distributed[41–45]. While we used pharmacology and chemogenetics to target CP-AMPArs and PV interneurons, respectively, it is possible that these methods could have off-target effects. For example, non-PV cells within NAc could express CP-AMPArs, and thus CP-AMPAr antagonism may act on other FSIs or non-PV cell types. Furthermore, it is possible that our targeting of PV

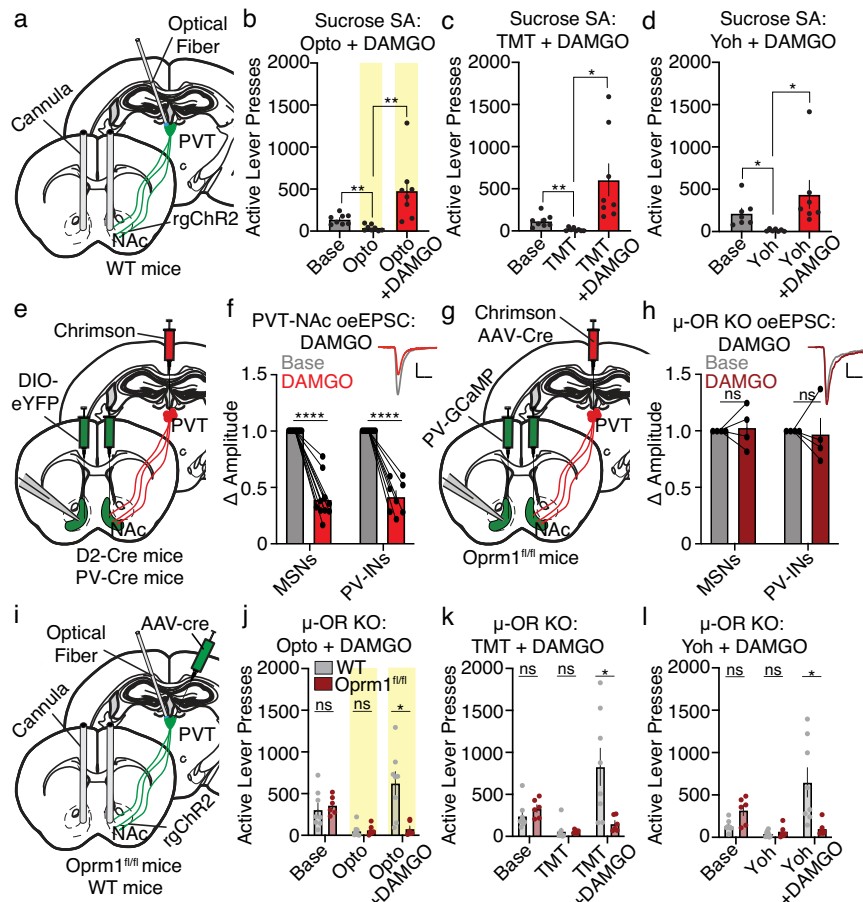

**Fig. 6 | PVT→NAc-dependent suppression of reward seeking is gated by presynaptic µ-opioid receptors. a** Surgical strategy for optogenetic manipulation of PVT→NAc neurons and intra-NAc DAMGO infusions. **b–d** Intra-NAc infusion of DAMGO prevented the suppression of sucrose self-administration caused by PVT→NAc stimulation (**b**), TMT (**c**), and yohimbine (**d**) (Opto: $n = 8$ mice/group; one-way ANOVA, $F_{2,21} = 9.33$, $P = 0.001$; planned two-tailed $t$-tests: base vs. opto $P < 0.01$, opto vs. DAMGO + opto $P < 0.01$; TMT: $n = 8$ mice/group; one-way ANOVA, $F_{2,21} = 8.02$, $P = 0.003$; planned $t$-tests: base vs. TMT $P < 0.01$, TMT vs. DAMGO + TMT $P = 0.02$; yohimbine: $n = 7$ mice/group; one-way ANOVA, $F_{2,18} = 4.26$, $P = 0.03$; planned $t$-tests: base vs. yohimbine $P = 0.02$, yohimbine vs. DAMGO + yohimbine $P < 0.05$). **e** Surgical strategy for patch-clamp electrophysiology. **f** Example electrophysiological waveforms (scale: 50pA/50 ms) and grouped data reveal that DAMGO reduced PVT→NAc^MSN and PVT→NAc^PV-IN oeEPSC amplitudes (MSNs: $n = 10$ cells, 6 mice; PV-INs: $n = 8$ cells, 5 mice; repeated-measures two-way ANOVA, effect of DAMGO: $F_{1,16} = 210.9$, $P < 0.0001$; Sidak's post-hoc: $P$-values < 0.001). **g** Surgical strategy for knockout of PVT µ-ORs with patch-clamp electrophysiology. **h** Example electrophysiological waveforms (scale: 50pA/50 ms) and grouped data reveal that

PVT µ-OR knockout prevented the DAMGO-induced decrease in PVT→NAc^MSN and PVT→NAc^PV-IN oeEPSC amplitudes ($n = 4$ cells per group, 2 mice; repeated-measures two-way ANOVA, effect of DAMGO $F_{1,6} = 0.0001$ $P = 0.99$). **i** Surgical strategy for PVT µ-OR knockout with simultaneous optogenetic manipulation of PVT→NAc neurons and intra-NAc DAMGO infusions. **j–l** Knockout of PVT µ-ORs in *Oprm1^fl/fl* mice rescued the suppression of sucrose self-administration caused by optogenetic stimulation of PVT→NAc neurons (**j**), TMT (**k**), and yohimbine (**l**) despite intra-NAc DAMGO infusions (Opto: $n = 6$ *Oprm1^fl/fl*, 8 WT mice; repeated-measures two-way ANOVA, day × group interaction: $F_{2,24} = 9.74$, $P = 0.001$; Sidak's post-hoc: opto + DAMGO $P = 0.02$; TMT: $n = 6$ *Oprm1^fl/fl*, 8 WT mice; repeated-measures two-way ANOVA, day × group interaction: $F_{2,24} = 7.04$, $P = 0.004$; Sidak's post-hoc: TMT + DAMGO $P < 0.001$; yohimbine: $n = 6$ *Oprm1^fl/fl*, 8 WT mice; repeated-measures two-way ANOVA, day × group interaction: $F_{2,24} = 10.97$, $P = 0.0004$; Sidak's post-hoc: yohimbine + DAMGO $P < 0.05$). µ-OR µ-opioid receptor, Base Baseline, Opto optogenetics, Yoh Yohimbine, KO knockout; Group comparisons: *$P < 0.05$, **$P < 0.01$, ****$P < 0.001$. Bar graphs are presented as mean values ± SEM. Source data are provided as a Source Data file.

interneurons could have profound effects on downstream neurophysiology, and therefore may not be completely selective for our circuit-of-interest. Finally, our PV-interneuron cell targeting is likely to select for only a subpopulation of PV-expressing neurons due to incomplete genetic penetrance of the PV-Cre transgenic mouse line[46,47]. Notably, our electrophysiological recordings suggest that our Cre-dependent targeting of PV interneurons at least selects for FSIs, as fluorescent cells within PV-Cre mice displayed fast-spiking properties. Additionally, we find that these cells are inwardly rectifying, suggesting the presence of CP-AMPARs. Despite these findings, future studies selectively targeting CP-AMPARs at PVT→NAc^PV-IN synapses could elucidate the precise role of these receptors for PVT→NAc-dependent behavioral suppression.

Here we find that the PVT→NAc brake for reward seeking is physiologically and functionally disrupted by systemic or local

opioid exposure. Consistently, previous studies reveal that accumbal µ-ORs can promote feeding[48–50], and can facilitate drug-seeking behaviors[51–53]. Our findings suggest that these behavioral effects could be, at least in part, related to the inhibition of PVT→NAc circuitry resulting in the disinhibition of reward-motivated behavioral actions. Despite our findings that PVT µ-OR knockout prevents systemic heroin or intra-NAc DAMGO infusions from disinhibiting sucrose-seeking behaviors, caveats regarding the specificity of our results should be considered. First, we cannot dissociate whether opioid-driven µ-OR activation on PVT somata or PVT→NAc axon terminals are required for our observed behavioral effects. Considering that heroin and DAMGO can dramatically reduce activity at both PVT→NAc somata and downstream synapses, it is possible that both mechanisms are involved. Second, additional mechanisms could also contribute to opioid-induced behavioral disinhibition,

such as μ-OR activation elsewhere in the brain and in other NAc cell types that express μ-ORs[54–56]. Overall, while our data suggests that opioid-induced inhibition of PVT→NAc neurons can disinhibit maladaptive behavioral actions, whether these effects are isolated to PVT→NAc somata and/or synapses has yet to be established.

Our data suggest that the PVT→NAc circuit is susceptible to opioid-induced plasticity; however, studies evaluating the function of PVT→NAc neuronal activity for opioid seeking in the self-administration paradigm have remained mixed. A recent study showed that stimulation of this pathway can modestly increase opioid seeking before, but not after, extinction learning[39]. In contrast, others showed that activation of this pathway can increase opioid seeking, but only under conditions of chronic food restriction[57]. Considering the ability of opioids to disengage PVT→NAc circuitry, it is possible that chronic opioid experience causes long-lasting adaptations at PVT→NAc synapses that could disrupt the ability of this circuit to suppress operant behaviors including opioid seeking. Future studies that evaluate cell-type-specific adaptations within PVT→NAc to PV-interneuron circuitry following opioid use are therefore warranted.

Our two-photon calcium imaging data highlights the heterogeneous, although overall inhibitory, PVT→NAc neuronal ensemble dynamics during sucrose self-administration and seeking[6,58]. Indeed, though we cluster PVT→NAc neurons by activity (see Fig. 1), each ensemble seems to contain subsets of cells with distinct responses, despite isolating PVT neurons by location (i.e., posterior) and connection (i.e., projections to NAc). Previous research has shown that unique PVT cell types can have opposing effects on behavior, as this structure contains many classes of neurons that can be distinguished by anatomical location (anterior versus posterior)[59–62], function[9,63,64], and gene expression[58,60]. While we find that PVT→NAc neuronal activity is modulated by μ-OR activity, there are an abundance of genotypic differences that have yet to be explored; moreover, whether these genotypic differences contribute to the heterogenous activity in PVT→NAc neuronal ensembles remains unclear. Future studies including high-throughput single-cell sequencing or targeting of unique cell types would aid in elucidating how these genetically distinct cell populations contribute to reward-seeking behavior.

Overall, we discover a keystone neuronal system for behavioral suppression and show that this system is rapidly disengaged by opioids. These findings provide a neuronal substrate whereby prescription or recreational opioid could lead to maladaptive behavioral actions and addiction. Newly synthesized opioids or competing pharmacologies that prevent the inhibition of thalamostriatal circuits could therefore reduce a patient's vulnerability to addiction.

## Methods
### Animals
All experiments were approved by the Institutional Animal Care and Use Committee (IACUC) at the Medical University of South Carolina in accordance with the NIH-adopted Guide for the Care and Use of Laboratory Animals (Protocol 2021-01835). Adult (8–24 weeks) male and female C57BL6/J wild-type (WT), PV-Cre (B6.Cg-Pvalbtm1.1(cre)Aibs/J, Strain #012358)[65], and D2-Cre (Drd2, line ER44, RRID:MMRRC_017263-UCD)[66] mice were used for experiments not involving genetic knockout of μ-ORs. B6129SF2/J WT (Jax Strain #101045) and Oprm1fl/fl (B6.129-Oprm1tm1.1Cgrf/KffJ, Jax Strain #030074)[37] mice were used for experiments involving genetic knockout of μ-ORs. All mice were group-housed pre-operatively and single-housed post-operatively, with access to standard chow and water ad libitum throughout all experiments (mice were at least 8 weeks of age and 20 g prior to study onset). Mice were housed under a reverse 12:12 h light cycle (lights off at 8:00am), with experiments performed during the dark phase.

### Surgery
For cranial surgeries, all mice were anesthetized with isoflurane (0.8–1.5% in oxygen; 1 L/minute) and placed within a stereotactic frame (Kopf Instruments). Ophthalmic ointment (Akorn), topical anesthetic (2% Lidocaine; Akorn), analgesic (Ketorolac, 2 mg/kg, ip), and subcutaneous sterile saline (0.9% NaCl in water) were given pre- and intra-operatively for health and pain management. An antibiotic (Cefazolin, 200 mg/kg, sc) was given post-operatively to reduce the possibility of infection, and mice were allowed to recover for at least 3 weeks after surgeries.

**Two-photon calcium imaging.** We targeted PVT→NAc projection neurons for two-photon calcium imaging through a single micro-injection of a Cre-inducible virus encoding the calcium indicator GCaMP6m (AAVdj-ef1a-DIO-GCaMP6m; 300 nl) into the posterior PVT (AP: −1.55 mm; ML: −1.13 mm; DV: −3.30 mm; 20° angle), along with bilateral microinjections of a retrogradely trafficked virus encoding Cre-recombinase (rgAAV2-hSyn-Cre; 500 nl) aimed for the anterior/medial NAc shell (AP: + 1.45 mm; ML: ± 0.65 mm; DV: −4.65 mm) in WT mice. A microendoscopic gradient refractive index lens (GRIN lens; 8 mm long, 0.5 mm diameter, Inscopix) was implanted dorsal to the PVT injection site (AP: −1.55 mm; ML: −1.13 mm; DV: −3.00 mm; 20° angle), allowing chronic visual access to PVT→NAc projection neurons[6,67]. A stainless-steel head ring was cemented around the GRIN lens using dental cement and skull screws for subsequent head fixation[26]. Histology confirmed that GRIN lens placements and GCaMP6m fluorescence allowed localized visualization of posterior PVT neurons post-mortem.

**Behavioral optogenetics, chemogenetics, and neuropharmacology.** We targeted PVT→NAc projection neurons for optogenetic manipulations using injection coordinates and volumes that were identical to two-photon calcium imaging experiments described above. Our first strategy was to give a single microinjection of a Cre-inducible virus encoding one of the three opsins (AAV5-ef1a-DIO-ChR2-eYFP; AAV5-ef1a-DIO-eNpHR3.0-eYFP; AAV5-ef1a-DIO-eYFP) into the posterior PVT, along with bilateral microinjections of the retrogradely trafficked virus encoding Cre-recombinase into the NAc shell (rgAAV2-hSyn-Cre) of WT mice. When combining these experiments with chemogenetics, we instead injected a retrogradely trafficked virus encoding channelrhodopsin (rgAAV2-hSyn-ChR2-eYFP) bilaterally into NAc shell, along with bilateral microinjections of a Cre-inducible virus encoding an inhibitory DREADD[68,69] into the NAc shell (AAV5-hSyn-DIO-hM4D(Gi)-mCherry) of PV-Cre or WT control mice. We also used this surgical strategy for μ-OR knockout experiments, wherein mice received a retrogradely trafficked virus encoding channelrhodopsin (rgAAV2-hSyn-ChR2-eYFP) bilaterally in NAc shell, and an injection of a virus encoding Cre-recombinase (AAV5-hSyn-Cre-eYFP) into PVT. For all optogenetics experiments, a custom-made optical fiber[70] was implanted dorsal to the PVT injection site (AP: −1.55 mm; ML: −1.13 mm; DV: −3.00 mm; 20° angle), allowing laser-evoked perturbation of activity in PVT→NAc projection neurons. For optogenetics experiments involving simultaneous chemogenetics or neuropharmacology, a bilateral double-barrel guide cannula (Plastics One: 26-gauge, 5 mm length, 1.2 mm barrel separation) was implanted dorsal to the anterior/medial NAc shell (AP: + 1.45 mm; ML: ± 0.60 mm; DV: −4.15 mm). A stainless-steel head ring was cemented around the optical fiber and/or guide cannula using dental cement and skull screws. Histology confirmed optical fiber placements and structure-localized fluorescence post-mortem.

**Immunohistochemistry and slice electrophysiology.** We targeted PVT→NAc projection neurons for both immunohistochemical and slice electrophysiological studies using injection coordinates and volumes that were identical to two-photon calcium imaging experiments described above. For immunohistochemistry, we gave a single

microinjection of a Cre-inducible virus encoding eYFP (AAV5-ef1a-DIO-eYFP) into the posterior PVT, along with bilateral microinjections of a retrogradely trafficked virus encoding Cre-recombinase into the NAc shell (rgAAV2-hSyn-Cre) of WT or $Oprm1^{fl/fl}$ mice. For slice electrophysiology of downstream targets, we gave a single microinjection of a virus encoding a red-shifted excitatory opsin (AAV5-hSyn-ChrimsonR-tdTomato)[31] into the PVT, along with bilateral microinjections of a Cre-inducible virus encoding eYFP into the NAc shell (AAV5-ef1a-DIO-eYFP) of D2-Cre or PV-Cre mice. For electrophysiology with PVT μ-OR knockout, we used a similar surgical strategy wherein we combined an AAV-Cre (AAV5-hSyn-Cre-eYFP) with AAV-ChrimsonR (AAV5-hSyn-ChrimsonR-tdTomato)[31] at a 1:10 ratio. For identification of PV-INs, we gave bilateral NAc shell microinjections of a PV-interneuron targeted calcium indicator (pAAV-S5E2-GCaMP6f)[38], which expresses GCaMP6f under control of the E2 regulatory element.

### Head-fixed behavior

**Self-administration.** Experiments involving sucrose self-administration were performed based on a previous study wherein we developed a model of natural- and drug-reward seeking in head-restrained mice, enabling simultaneous two-photon calcium imaging[71]. After recovery from surgery, mice were habituated to head fixation during 30 min sessions wherein levers were not presented.

**Acquisition.** Mice next acquired sucrose self-administration through 14 daily 1 h sessions, during which two levers and a sucrose delivery spout were placed within forelimb reach. Sucrose delivery spouts were arranged equally between both levers so that mice would not be biased toward either lever. A press on the active lever, but not inactive lever, resulted in the presentation of a tone cue (8 kHz, 1.6 s) followed by the delivery of a liquid sucrose reward (12.5 μl; 12.5% mixed in tap water; 2 s delivery duration). A timeout period (20 s) was given after each cue- and sucrose-reinforced active lever press, wherein active lever pressing had no effect. Mice were capped at 10 sucrose droplets on days 1–2 of acquisition, 20 sucrose droplets on days 3–4 of acquisition, and 40 sucrose droplets on days 5–14 of acquisition to match previous drug self-administration experiments (capping prevents drug overdose in those studies).

**Suppression.** Following acquisition of lever pressing for sucrose, we used several methods to suppress active lever pressing behavior. We first used a predator odor to suppress sucrose self-administration, wherein mice were exposed to the fox feces derivative TMT (30 μL; 1% v/v ddH2O) or vehicle (order counterbalanced) within their home cage for 15 min[72]. Mice were removed from their home cage after TMT or vehicle exposure, and immediately underwent a normal, uncapped 45 min sucrose self-administration session. We also used a pharmacological stressor to suppress sucrose self-administration, wherein mice were given a systemic injection of yohimbine (0.625 mg/kg, i.p.; Sigma Chemical)[72] or vehicle (order counterbalanced) 15 min before a normal, uncapped 45 min sucrose self-administration session. Finally, we used extinction learning to suppress active lever pressing. Following at least 2 normal acquisition sessions (no stressors), mice were given daily 1 h sessions wherein active lever pressing no longer resulted in cue or sucrose delivery until extinction criteria were reached. Extinction criteria were determined a priori[71], as (1) at least 10 days of extinction training and (2) 2 of the last 3 days at a maximum of 20% active lever pressing rate as compared with the last 2 days of acquisition.

**Reinstatement.** Following extinction learning, mice were given a cue-induced reinstatement test wherein active lever pressing resulted in tone cue presentation (as in acquisition), but not sucrose delivery. A timeout period (20 s) was given after the onset of each cue, wherein active lever pressing did not result in cue delivery. Graphical

representations of these as data as bar graphs refer to the mean and standard error of the mean (SEM).

### Real-time place preference

Real-time place preference was conducted using a standard 2-chamber conditioning apparatus that was partially divided by a partition which created two visually (white versus black walls) and texturally (rod versus grate flooring) distinct environments. Optogenetic manipulation of PVT→NAc neurons was performed in mice expressing ChR2 or eYFP. Real-time place preference was assessed over two consecutive testing sessions (one 15 min session per day), where optogenetic light was counterbalanced between environments for each mouse and session. On the first testing day, eYFP and ChR2 mice received an intraperitoneal injection of saline immediately before being placed in between the two chambers. On the second testing day, eYFP and ChR2 mice received an intraperitoneal injection of heroin (1 mg/kg; i.p.) immediately before being placed in between the two chambers. Place preference data was analyzed by comparing the time in each chamber across groups, using two-way ANOVAs followed by Sidak's post-hoc analyses. Paired t-tests were used to compare chamber entries between groups (an index of locomotion). Graphical representations of these as data as bar graphs refer to the mean and standard error of the mean (SEM).

### Two-photon calcium imaging

**Data collection and processing.** We visualized GCaMP6m-expressing PVT→NAc projection neurons using a two-photon microscope (Bruker Nano Inc) equipped with a tunable InSight DeepSee laser (Spectra Physics, laser set to 920 nm, -100 fs pulse width), resonant scanning mirrors (-30 Hz framerate), a ×20 air objective (Olympus, LCPLN20XIR, 0.45NA, 8.3 mm working distance), and GaAsP photodetectors. In some cases, two fields of view (FOVs) were visible through the GRIN lens (separated by >75 μm in the Z-axis to avoid signal contamination from chromatic aberration), in which case we recorded from each FOV during separate imaging sessions. Data were acquired without averaging using PrarieView software, converted into hdf5 format, and motion corrected using SIMA[73]. Following motion correction, a motion-corrected video and averaged time-series frame were used to draw regions of interest (ROIs) around dynamic and visually distinct cells using the polygon selection tool in FIJI[74]. Fluorescent traces for each ROI were then extracted using SIMA, and all subsequent analyses were performed using custom Python codes in Jupyter Notebook[6,25]. Two-photon imaging was performed during select acquisition sessions (early: days 1–2; middle: days 7–8; late: days 13–14) and extinction sessions (early: days 1–2; late: last 2 days) to simplify data analysis.

**Data analysis.** We quantified the average or 'basal' fluorescence of each neuron across time, as basal fluorescence can serve as a proxy for firing rates in tonically active cell populations[25,26], including PVT→NAc neurons[6]. Fluorescent traces were averaged across 3 min bins and normalized to the first bin of each session. In the case of sucrose self-administration sessions, which were of varying length depending on the day and speed of intake, we compared first and last bin for each session. Data for each neuron was normalized to the baseline averaged across neurons, allowing assessment of within-session adaptations. These data were compared across time and sessions using a two-way ANOVA, followed by Sidak's post-hoc analyses for between-session comparisons.

In addition to fluorescence adaptations within sessions, we aligned fluorescent traces of each neuron to active lever presses, including the 10 s beforehand, 3 s between the lever press and sucrose delivery, and 10 s after sucrose delivery. The 23 s fluorescent trace was averaged across trials and plotted as a peri-stimulus time heatmap across neurons. Due to the robust active lever pressing rates both late

in acquisition and during cue-induced reinstatement, the resulting two-dimensional arrays from those sessions were used to inform separate principal components analyses. The number of principal components were determined using the inflection point of a scree plot, which graphs the peri-stimulus time histogram of variance explained versus an increasing number of principal components. The remaining principal components were then plotted into a subspace and used to inform the Scikit-learn function *sklearn.cluster.SpectralClustering*, a spectral-clustering algorithm that uses a k-nearest neighbor connectivity matrix to identify unique cell clusters. Spectral clustering was chosen due to its improved performance for separating dynamic neuronal datasets as compared with other clustering algorithms[24,25]. Finally, a decoding analysis was used to determine how activity of each neuron could predict future active lever pressing behavior. A binary decoder was used through the Scikit-learn functions *sklearn.discriminant_analysis, sklearn.smv*, and *sklearn.decomposition*, informed by the fluorescence of each neuron during 2 epochs: 1 s before each active lever press vs a random 1 s baseline. As a control, these 2 epochs were randomly shuffled, and the decoding analysis was repeated. We used a 1-second epoch prior to the lever press based on: (1) a pre-lever press epoch would ensure that the decoding was not due to cue presentation, liquid delivery, or sucrose consumption and (2) past studies have demonstrated single-cell calcium events during a 1-second pre-reward trace interval can be used to accurately predict reward learning within a Pavlovian conditioning task[6,26]. Thus, a 1 s epoch immediately before lever pressing seemed most appropriate for our decoding analysis and was chosen beforehand such that only one analysis was performed. The decoding scores for each neuron was subtracted from the average of shuffled data for the corresponding group of cells (chosen by day and/or ensemble), and the data was plotted against the shuffled data for all neurons. These data were compared across ensembles and corresponding shuffled data using a two-way ANOVA, followed by Sidak's post-hoc comparisons.

Subsets of neurons were reliably identified across days based on structure and relative position within each FOV, allowing us to visualize the adaptation and maintenance of responses from individual neurons across specific timepoints. Single-cell tracking was performed at two timepoints: (1) across early to late sucrose self-administration (see Supplementary Fig. 2); (2) across saline and heroin injection test days (see Fig. 4). Next, the activity of tracked cells could be analyzed across days using the population decoding, clustering, and ensemble decoding analyses described above. In addition to these analyses, we performed a response adaptation analysis wherein the mean response of each neuron was compared across days and ensembles. These data were analyzed using a two-way ANOVA, followed by Sidak's post-hoc comparisons. Lastly, a Pearson's correlation analysis was used to determine the linear association of responses across tracked acquisition sessions. Graphical representations of these as data as bar graphs refer to the mean and standard error of the mean (SEM).

## Behavioral optogenetics, chemogenetics, and neuropharmacology

**Optogenetics.** We used optogenetics to stimulate or inhibit the activity of PVT→NAc neurons during the expression of sucrose taking (late acquisition) and seeking (cue-induced reinstatement), and during the suppression of sucrose taking (TMT or yohimbine tests) and seeking (late extinction) as described above. For photoactivation experiments in ChR2 or control eYFP mice, the laser (473 nm; ~10 mW) was pulsed (5 ms; 20 Hz) every for 30 s intervals once/minute throughout the session. For photoinhibition experiments in eNpHR3.0 or control eYFP mice, the laser (532 nm; ~10 mW) was displayed (pure light, not pulsed) for 30 s intervals once/minute throughout the session. Because each laser manipulation had no effect in control eYFP mice, these data were collapsed across groups for each experiment.

**Chemogenetics and neuropharmacology.** We used site-specific chemogenetics and neuropharmacology to determine the function of NAc cell types and signaling mechanisms for the suppression of sucrose taking and seeking. For chemogenetics experiments, microinfusions of saline vehicle (0.9% NaCl in water; 0.3 µL/side) or CNO (0.1 µg in 0.3 µL/side) were administered into NAc 5 min before each behavioral session (session order counterbalanced). For neuropharmacology experiments, microinfusions of the vehicle saline (0.3 µL/side), D1-receptor antagonist SCH-23390 (0.6 µg in 0.3 µL/side)[75], D2-receptor antagonist raclopride (3 µg in 0.3 µL/side)[75], CP-AMPAr antagonist IEM-1640 (0.3 µg in 0.3 µL/side)[32], or the µ-opioid receptor agonist DAMGO (0.02 µg in 0.3 µL/side)[52] were administered 5 min before each behavioral session (saline and antagonist session order counterbalanced).

**Heroin experiments.** We tested the influence of systemically administered heroin (1 mg/kg; i.p.) on PVT→NAc neuronal activity and on behavior. Heroin was administered 5 min after the onset of two-photon imaging experiments, such that within-session adaptations could be evaluated. For behavioral experiments, heroin was administered immediately before sucrose self-administration or real-time place preference testing (note that sucrose self-administration was also coupled with two-photon imaging). Behavioral data were analyzed across groups and/or behavioral sessions using ANOVA, followed by Sidak's post-hoc comparisons when applicable. Graphical representations of these as data as bar graphs refer to the mean and standard error of the mean (SEM).

## Patch-clamp electrophysiology

Mice were euthanized between 4 and 6 weeks following surgery. At time of euthanasia, mice were deeply anesthetized with isoflurane (1.5% in oxygen; 1 L/min) before transcardial perfusion with oxygenated, ice-cold sucrose-based cutting solution containing the following (in mM): 225 sucrose, 119.0 NaCl, 1.0 NaH$_2$PO$_4$, 4.9 MgCl$_2$, 0.1 CaCl$_2$, 26.2 NaHCO$_3$, 1.25 glucose (305–310 mOsm). Brains were rapidly removed and bathed in cutting solution, while coronal sections 300 µm thick were taken using a vibratome (Leica VT1200S). Sections were incubated in warm aCSF (32 °C) containing the following (in mM): 119 NaCl, 2.5 KCl, 1.0 NaH$_2$PO$_4$, 1.3 MgCl, 2.5 CaCl$_2$, 26.2 NaHCO$_3$, 15 glucose (305–310 mOsm). After at least 1 h of recovery, slices were constantly perfused with aCSF and visualized using differential interference contrast through a ×40 water-immersion objective mounted on an upright microscope (Olympus BX51). Whole-cell patch-clamp recordings were obtained using borosilicate pipettes (~3–5 MΩ) backfilled with a potassium gluconate-based internal solution composed of the following (in mM): 130 K-gluconate, 10 KCl, 10 HEPES, 10 EGTA, 2 MgCl$_2$, 2 ATP, 0.2 GTP (pH 7.35, mOsm 280) to characterize action potential waveforms and to measure the amplitude of PVT→NAc synaptic currents (mediated by AMPA receptors). Alternatively, recordings were obtained using a cesium methylsulfonate-based internal solution composed of the following (in mM): 117 Cs methanesulfonic acid, 20 HEPES, 2.8 NaCl, 5 TEA, 2 ATP, 0.2 GTP (pH 7.35, mOsm 280) to measure the amplitude of synaptic currents and AMPA rectification.

**NAc electrophysiology.** Patch-clamp recordings were obtained from eYFP$^+$ (PV-INs in PV-Cre and D2-MSNs in D2-Cre mice) and eYFP$^-$ (putative D1-MSNs in D2-Cre mice) neurons surrounding the virus injection site in anterior/medial NAc shell. eYFP was visualized using a blue LED (<1 mW) and a GFP epifluorescence filter set. In a subset of accumbal neurons, depolarizing current pulses (800 ms; 50 pA steps) were applied in current-clamp mode to confirm that recordings were from fluorescence-identified cell types. Specifically, PV-INs were confirmed based on their fast-spiking properties, whereas D1- and D2-MSNs were confirmed by their relatively limited spike frequency,

ramping depolarization, and/or late spiking features. We also evaluated functional synaptic innervation from PVT to each of the NAc cell types. Visually identified NAc neurons were held at −70mV in voltage-clamp mode, and presynaptic ChrimsonR-tdT$^+$ axons from PVT were activated using a green LED (10 ms pulse, 1 mW) pulsed every 10–15 s. The peak amplitude of reliably evoked excitatory postsynaptic currents was measured for each neuron and compared across cell types using a one-way ANOVA followed by Sidak's post-hoc tests for between group comparisons. In a subset of recordings, we confirmed that optically evoked excitatory postsynaptic currents (oeEPSCs) were blocked by 10 min bath application of the glutamatergic AMPA receptor antagonist DNQX (10 μM). Additionally, we examined the influence of the CP-AMPAr antagonist IEM-1640 (50 μM)[32] and μ-opioid receptor antagonist DAMGO (3 μM)[76] on oeEPSC amplitudes through bath application for 25 min. For these pharmacological experiments, the average amplitude of the AMPA receptor-mediated oeEPSCs was taken from the first 5 min of recordings (before drug application) and the last 5 min of recordings (after drug application). Responses were compared across time and cell types using a two-way ANOVA, followed by Sidak's post-hoc tests for between group comparisons. AMPA rectification was measured using the cesium-based internal solution in voltage-clamp mode, with a GABA receptor antagonist picrotoxin (100 μM)[13] and the NMDA receptor antagonist APV included in the perfusion aCSF (50 μM)[13]. oeEPSCs were taken as above, with 5 sweeps averaged at a range of voltages (−80, −70, −50, −30, −10, +10, +30, +50 mV). Data were normalized to the peak oeEPSC at −80mV, and an AMPA rectification index was calculated for each neuron as $I_{+50mV}/I_{-70mV}$ where I is the peak oeEPSC amplitude at each voltage. AMPA rectification was then compared across neurons using a one-way ANOVA, followed by Sidak's post-hoc tests for between group comparisons.

**NAc electrophysiology with PVT μ-OR knockout.** In *Oprm1$^{fl/fl}$* mice, patch-clamp recordings were obtained from GCaMP6f$^+$(PV-INs) and GCaMP6f$^-$ (putative MSNs) neurons surrounding the virus injection site in anterior/medial NAc shell. Similar to above, GCaMP6f$^+$ neurons were visualized using a blue LED (<1 mW) and a GFP epifluorescence filter set. We examined the influence of DAMGO (3 μM)[76] on oeEPSC amplitudes through bath application for 25 min, wherein the average amplitude of the AMPA receptor-mediated oeEPSCs was taken from the first 5 minutes of recordings (before drug application) and the last 5 min of recordings (after drug application). Responses were compared across time and cell types using a two-way ANOVA, followed by Sidak's post-hoc tests when applicable.

**PVT electrophysiology.** Patch-clamp recordings were obtained from eYFP$^+$ neurons in the posterior PVT of Cre-infused *Oprm1$^{fl/fl}$* mice, or from eYFP$^-$ cells in naïve *Oprm1$^{fl/fl}$* mice. In current-clamp mode, cells were held at −70 mV and depolarizing current pulses (800 ms; 10–300 pA) were applied across 15 ten-second sweeps in attempt to evoke 4–10 action potentials per sweep. Baseline recordings were taken directly before drug application, whereas 'DAMGO' recordings were taken immediately following 25 min of DAMGO (3 μM)[76] bath application. To calculate change in neuronal spiking for each cell, post-wash spiking was normalized to the average number of spikes from the baseline recording. Data from each group were compared across time using paired *t*-tests.

**DIO-Gi-DREADD validation.** Patch-clamp recordings were obtained from mCherry$^+$ or neighboring mCherry$^-$ neurons in the anterior/medial NAc shell of PV-Cre mice. In current-clamp mode, cells were held at −70mV and depolarizing current pulses (800 ms; 10–300pA) were applied across 15 ten-second sweeps in attempt to evoke 4–10 action potentials per sweep. Baseline recordings were taken directly before drug application, whereas post-CNO

recordings were taken following 5-minutes of CNO (5 μM)[68] bath application. To calculate change in neuronal spiking for each cell, post-wash spiking was normalized to the average number of spikes from the baseline recording. Data was compared across time and cell types using a two-way ANOVA, followed by Sidak's post-hoc tests for between group comparisons. Graphical representations of these as data as bar graphs refer to the mean and standard error of the mean (SEM).

**Immunohistochemistry**
Free-floating 80 μm coronal sections containing the PVT or NAc were blocked in 0.1 M PBS with 2% Triton X-100 (PBST) with 2% normal goat serum (NGS, Jackson Immuno Research, Westgrove, PA) for 2 h at room temperature with agitation. Sections were then incubated overnight at 4 °C with agitation in the appropriate primary antisera (see Supplementary Table 1) diluted in 2% PBST with 2% NGS, washed three times for 5 min in PBST, then incubated in the appropriate secondary antisera diluted in PBST with 2% NGS for 4 h at room temperature with agitation. All secondary antisera were raised in goat, conjugated to Alexa fluorophores, were used at a concentration of 1:1000, and were purchased from Invitrogen (Carlsbad, CA). Sections were then washed three times for 5 min in PBST, mounted on Super-Frost+ slides, and cover slipped with ProLong™ Gold Antifade. Slides were stored in a dark area. Brain sections were imaged using a Leica SP8 laser-scanning confocal microscope. Care was taken to only acquire images within dense fields of virally transduced neurons (eYFP$^+$) in the NAc. Simultaneously, we also imaged immunohistochemically (IHC) labeled markers for NeuN, pre-pro Enkephalin (ppENK), nNOS, Parvalbumin (PV), Choline acetyltransferase (ChAT), or μ-opioid receptors (μ-ORs). Each non-eYFP signal was processed in separate IHC runs. For detection of eYFP$^+$ cells, an OPSL 488 nm laser line was used. For all other cell-type markers, a Diode 638 nm laser line was used for detection. During all imaging experimentation, a frame size of 1024 × 1024 was used and pinhole size, laser power, gain, and Z-step thickness were held constant throughout. For cell-type-specific staining IHC and eYFP$^+$ cell counting experimentation, a ×20 air objective was used. In both imaging modalities Z-step thickness was derived using Nyquist parameters to obtain optimal sampling for our quantitative analyses.

All image analyses were performed on 3D reconstructed Z-series datasets. Once deconvolved (Hyugens deconvolution, SVI) datasets were imported to Imaris (v 9.0, Bitplane) for cell counting. The "spot" tool in Imaris was used in a semi-automated manner to detect and quantify virally labeled eYFP+ cells as well as immunohistochemically labeled nNOS$^+$, ppENK$^+$, PV$^+$, or ChAT$^+$ cells. Next, Imaris was used to quantify the number of cells with coincident eYFP and, nNOS, ppENK, PV, or ChAT labeling. For each cell type investigated, raw cell counts were expressed relative to dataset volume (μm$^3$), to create a cell/μm$^3$ index and to account for potential variation in tissue thickness. To determine the cellular identity of NAc neurons that receive PVT input, the number of eYFP$^+$ cells that were coincidently ppENK$^+$, PV$^+$, ChAT$^+$ or nNOS$^+$ were calculated and expressed as a percentage of the total eYFP$^+$ population within each field imaged, again using 3–5 datasets per animal, sampling each hemisphere, across 4 animals. With these data, we generated percentages of D2 (ppENK$^+$), PV, ChAT, and nNOS neurons in each field that receive PVT input (identified via eYFP expression). Given the known population density of D1 and D2-MSNs at ~95% of total cells and the relatively small population density of interneurons within the NAc, we summed each cell-type average percentage and inferred that the remaining fraction of eYFP$^+$ NAc cells that did not show reactivity for ppENK, PV, ChAT or nNOS were putative D1-MSNs. For statistical analysis, IBM SPSS (version 24) was used to conduct comparisons between cell type across region via one-way ANOVAs, followed by planned contrasts for each subregion of interest (NAc

core vs NAc shell), with PVT→NAc PV⁺ neuron counts being serially compared to each additional marker (nNOS, ppENK, ChAT). For all planned contrasts, equal variances were not assumed. Graphical representations of these as data as bar graphs refer to the mean and standard error of the mean (SEM).

### RNAscope

**Data collection.** Six mice (3 WT, 3 *Oprm1*^fl/fl^) were deeply anesthetized before euthanasia. Following decapitation, brains were extracted and immediately submerged in 2-methyl-butane on dry ice (30 s at −40°F). Tissue was stored at −20 °C until 20 μm coronal sections were cut in a Leica cryostat (one WT brain was lost during sectioning) and mounted with a brush onto room temperature Superfrost Plus slides (Fisher Scientific). Slides were stored at −80 °C until processing a day later. The slides were processed with RNAscope probes and reagents according to the RNAscope Multiplex Fluorescent v2 Assay manual (Advanced Cell Diagnostics). To determine *Oprm1* mRNA expression in PVT neurons, three slides per mouse brain with 2–4 adjacent sections through the posterior PVT (AP: −1.2 to −2.2 mm) were fixed in 4% PFA for 15 min, then twice rinsed in PBS. Sections were dehydrated sequentially in 50%, 70%, and 100% ethanol. A hydrophobic barrier was drawn around the sections to limit spread of solutions on each slide. Using an RNAscope Reagent Kit, the slides were incubated in RNAscope Hydrogen Peroxide (10 min), rinsed twice in dH₂0, and then treated with Mm-*Oprm1* (Cat # 315841) and Mm-*Oprk1* (Cat # 316111-C2) probes for 2 h at 40 °C in an ACD hybridization oven. The sections were incubated with amplification (AMP) solution 1 (30 min, 40 °C), 2 (15 min, 40 °C), and 3 (30 min, 40 °C) followed by a double rinse in 1× RNAscope Wash Buffer (2 min) between each amplification step. The slides were then incubated with fluorescently labeled probes (Akoya Bio) to distinguish each channel TSA + Cyanine 3 (Cat t# NEL744001KT) and TSA + Cyanine 5 (Cat # NEL745001KT). Next, a drop of DAPI (30 s) was added to each slide, followed by Prolong Gold fluorescent mounting medium (Invitrogen) and a coverslip.

**Confocal imaging and quantification.** Z-series digital images of each section were taken on a Leica SP8 laser-scanning confocal microscope with a ×10 air objective using 2X digital zoom. YFP expression was captured with the 488 nm laser line, Cy3 was captured with the 552 laser line, and Cy5 was captured with the 638 laser line. All imaging parameters (i.e., optical zoom, field size, laser power, gain and pinhole) were kept consistent between WT and *Oprm1*^fl/fl^ mice. Following acquisition, datasets were imported into Imaris 9.0 (Bitplane). Within Imaris, the "spots" tool was used to mark and identify individual cells by virtue of κ-OR expression within a field of YFP expression. Within each identified cell (marked by its corresponding spot region), both κ-OR and μ-OR sum intensity values were collected for both genotypes. Approximately 130–200 cells were identified within each optical dataset. Nine optical datasets were collected for each genotype, across 3 *Oprm1*^fl/fl^ and 2 WT mice. μ-OR expression values were expressed relative to κ-OR, with the total average κ-OR values for either WT or *Oprm1*^fl/fl^ being set to 100% and WT or *Oprm1*^fl/^ κ-OR or μ-OR values being expressed relative its corresponding κ-OR average. Changes in μ-OR or κ-OR expression were analyzed across groups using a two-way ANOVA, followed by a Sidak's post-hoc comparison. Graphical representations of these data as bar graphs refer to the mean and standard error of the mean (SEM).

### Statistics and reproducibility

We used Graphpad PRISM (v8) for behavioral and electrophysiological data analysis, Python (v2.7) for imaging data analysis, and Adobe Illustrator (v26) for creating figures. Statistical comparisons were made using two-tailed paired or unpaired Student's *t*-tests and analysis of variance (ANOVA) followed by appropriate post-hoc comparisons, as indicated for each dataset. Sample sizes were based on previous

studies and/or estimated power analyses with an alpha significance value of 0.05 and power estimate of 0.8. All bar and line graphs reported in the figures are presented as mean values ± SEM, and individual dots within bar graphs refer to independent samples. Sample images for histology experiments are shown within the manuscript and were confirmed for each mouse within the experiment. Brain outlines used for histological representation of virus injections (e.g., Figs. 1i, 2a) were adapted from a previous article[77]. This article was published in The Mouse Brain in Stereotaxic Coordinates, Paxinos and Franklin, Figure 22 and 43, Copyright Elsevier (2004).

### Reporting summary

Further information on research design is available in the Nature Portfolio Reporting Summary linked to this article.

## Data availability

The behavioral and electrophysiological data generated in this study have been deposited in the Otis Lab GitHub database under accession code https://github.com/jimotis/Vollmer-et-al-2022-Nat-Comms. The behavioral and electrophysiological data generated in this study are also provided in the Supplementary Information/Source Data file. Two-photon imaging datasets, which are several terabytes, will be promptly made available upon request but are not immediately available for download due to file size. Source data are provided with this paper.

## Code availability

Code used for two-photon calcium imaging data analysis is available online: https://github.com/jimotis/Vollmer-et-al-2022-Nat-Comms.

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

## Acknowledgements

We would like to thank Maribel Vasquez-Silva for technical advice and support. This study was funded by grants from the National Institute of Drug Abuse (NIDA), F31-DA052186 (K.M.V.), F32-DA053830 (E.M.D.), R01-DA051650 and R01-DA054271 (J.M.O.), R01-DA054154 (M.D.S.), P50-DA046373 (J.F.M.), T32-DA007288 (R.I.G., J.E.P., A.R.D., K.M.V., E.M.D., B.M.S.), F32-DA050427 (B.M.S.), R25-GM113278 (P.N.S., K.T.W.), and the MUSC College of Medicine (COMETS; J.M.O. and M.D.S.).

## Author contributions

K.M.V., L.M.G., M.D.S., and J.M.O. designed the experiments and wrote the manuscript. K.V.M., L.M.G., R.I.G., K.T.W., E.M.D., C.W.B., J.E.P., R.E.C., A.T., P.N.S., B.B., B.M.S., A.R.D., A.M.W., T.C.J., J.A.R., J.F.M., M.D.S., and J.M.O. provided technical assistance and intellectual feedback on the project.

## Competing interests

The authors declare no competing interests.

## Additional information

Article

Kelsey M. Vollmer[1,4], Lisa M. Green[1,4], Roger I. Grant[1], Kion T. Winston[1], Elizabeth M. Doncheck[1], Christopher W. Bowen[1], Jacqueline E. Paniccia[1,2], Rachel E. Clarke[1,2], Annika Tiller[1], Preston N. Siegler[1], Bogdan Bordieanu[1], Benjamin M. Siemsen [2], Adam R. Denton[1,2], Annaka M. Westphal[2], Thomas C. Jhou[1], Jennifer A. Rinker[1], Jacqueline F. McGinty[1], Michael D. Scofield[2] & James M. Otis[1,3] ✉

[1]Department of Neuroscience, Medical University of South Carolina, Charleston, SC 29425, USA. [2]Anesthesiology and Perioperative Medicine, Medical University of South Carolina, Charleston, SC 29425, USA. [3]Hollings Cancer Center, Medical University of South Carolina, Charleston, SC 29425, USA. [4]These authors contributed equally: Kelsey M. Vollmer, Lisa M. Green. ✉e-mail: otis@musc.edu

