## [Peer Review File · Nature Communications]

An opioid-gated thalamoaccumbal circuit for the suppression of reward seeking in miceREVIEWER COMMENTS

Reviewer #1 (Remarks to the Author):

The present study by Vollmer, Green et al. test the hypothesis that a thalamostriatal circuit inhibits reward seeking during exposure to fear-provoking stimuli and that opioid receptor activation promotes risky behavior in the face of these stimuli by inhibiting this circuit via a presynaptic site of action. First the authors demonstrate that thalamostriatal PVT cells display heterogenous calcium responses to lever press activity (type 1 and 3 responses) associated with sucrose administration, and some ensembles are inhibited in a tonic manner. The authors demonstrate that optogenetic activation of PVT-NAcc pathway inhibited sucrose self-administration, while optogenetic inhibition of thalamostriatal neurons reversed TMT, yohimbine, and extinction-mediated reductions in sucrose self-administration. The authors subsequently demonstrated that thalamostriatal neurons innervate MSNs and PV interneurons, with the latter containing CP-AMPA receptors. The authors then show that the effects of optogenetic activation of PVT to NAcc afferents on sucrose self-administration is blocked by antagonism of CP-AMPA receptors in the NAcc and chemogenetic inhibition of NAcc PV neurons. The authors claim that thalamostriatal neurons and their terminals in the NAcc express MOR and that acute heroin administration decreases ensemble decoding of sucrose self-administration and blocks the suppressive effects of PVT to NAcc pathway stimulation and suppression induced by fear provoking stimuli. Lastly, the authors demonstrate that intra-NAcc DAMGO injections inhibit optogenetic -, TMT- and yohimbine-induced decreases in sucrose self-administration. They show that synapses from PVT to NAcc PV neurons are inhibited by DAMGO and that MOR deletion from PVT blocks the ability of intra-NAcc DAMGO from reversing the inhibitory effects of thalamostriatal optogenetic stimulation on sucrose self-administration. Overall, the study is of importance and tackles an important question. The authors test their hypothesis using an impressive set of interdisciplinary approaches. However, the authors make strong conclusions about their data that are not entirely supported by the present findings. Specifically, there are significant gaps that need to be addressed for the conclusions in their present form to be drawn, which include additional experiments and/or dialing back some of the interpretations and discussing caveats and alternative hypotheses. Below are some comments that will aid the authors in strengthening their manuscript and improving the cohesiveness of the study.

Comments

- There is a disconnect between the data across the different figures. The authors make strong claims that opioids acting through presynaptic receptors disrupt PVT inhibition, but this is largely decoupled from recordings of cell bodies and the effect of heroin presented earlier in the paper. Opioid receptors on terminals would inhibit terminal release and any potential influence on cell body activity in thalamostriatal neurons would likely be an indirect consequence. That is, is there any relationship between what MORs are doing at terminals and what is happening in the cell bodies of thalamostriatal neurons? Moreover, it is unclear whether the exciting findings in Fig 5 are related to the findings with heroin. Additional studies linking the role of MORs on PVT terminals, activity of thalamostriatal cells, and how this is involved in heroin's effects would be needed to reach the interpretations presented, such as

deleting MOR from thalamostriatal neurons and determining if that impacts the effect of heroin.

- The authors should discuss the caveat that IEM-1640 may be acting on other cell types not examined in the present paper or that CP-AMPA receptors may become engaged downstream of PVT inputs to NAcc, including the PV neurons. The authors' interpretation should be much more cautious without additional evidence.
- Caveats associated with electrophysiological characterization of synaptic connections onto NAcc neurons should be discussed/addressed. A between subjects design comparing MSNs and PV neurons is subject to variability from ChR2 expression, and as such additional measures would be useful for determining synaptic strength differences between cell types. Moreover, since virus was used to label D2 cells the caveat that unlabeled cells may constitute putative D1 and unlabeled D2 cells should be addressed. That PV neurons display larger currents relative to MSNs is not surprising, since PV interneurons are broadly characterized by the presence of CP-AMPA receptors and enhanced excitatory synaptic strength in striatal structures and cortex. Furthermore, NAcc PV neurons have been implicated in mediating aversive behavior independent of the PVT (see work by Morales group (NIDA) for example).
- MOR activation may also inhibit glutamate release onto non-PV targets (e.g. MSNs), which may be contributing to the observed behavioral effects. The authors do not examine these synapses and make strong conclusions about PVT inputs to PV that warrant further investigation. Furthermore, their physiology does not directly demonstrate presynaptic effects with the data presented, though it's implied.
- In fig 1, The authors describe a tonic decrease in ensemble 2 and 3 activity over the course of the sucrose SA session that is blocked by fear provoking stimuli and extinction and reappears upon reinstatement. However, the emphasis is placed on ensemble 3 activity aligned to lever pressing based on decoding accuracy. Further discussion on this would be useful for the reader.
- The authors state that ensemble 1 and 3 decoding accuracy is decreased after heroin due to decreased changes in calcium activity, which is consistent with 4H. However, 4G suggest that decoding accuracy in ensemble 1 is increased relative to saline day. The same mismatch appears for ensemble 2, but not as pronounced as ensemble 1.
- The switch to real time place preference evoked by opto PVT to NAcc stim with heroin injection is very interesting and in some ways may be relevant to the potentiation of sucrose self-administration that is observed in response to opto stim, TMT, and yohimbine in fig 4 and 5. This suggests that MOR activity may be unmasking a population of pro-sucrose administration neurons upon inhibition of MOR sensitive neurons.
- In supplementary figure s2, cluster 1 seems to contain a subset of cells that are briefly inhibited prior to lever press and are excited during reward delivery late in acquisition and inhibited during

reinstatement, suggesting they are distinct from other cells in cluster 1. This cluster in some ways has both cluster 1 and 3 properties.

- The MOR staining of PVT neurons and their terminals could be due to chance alone as MOR-like immunoreactivity appears to be widespread across the image. In the zoom within the representative images it appears that background from adjacent regions was cropped or subtracted. Immunostaining for MOR with this in MOR loxP mice would be useful for validating the antibody and genetic approach.
- NAcc PV cells are sparse relative to other interneuron populations. Moreover, PV-Cre mice used in the present study have incomplete genetic penetrance in the NAcc and may select for subtypes of NAcc fast-spiking interneurons in addition to non-PV cells with CP-AMPA receptors (see Adam Carter's work and Yan Dong). The authors should mention the incomplete penetrance to let the reader know that this may pertain to a selective sub-population of PV cells and overall broaden the discussion of how PV cells are limited in density in NAcc relative to striatum in the context of the present work.
- The authors should show the lever press – aligned population activity during the fear provoking stimuli in addition to the change across the session that is reported.
- The intrinsic properties of the MSNs in the representative traces do not show the well-characterized regular spiking properties of healthy MSNs ex-vivo, raising the possibility the cells / preparation was not optimal.
- The authors should cite relevant work on thalamostriatal circuitry where appropriate from the Penzo group at NIMH describing a role for thalamostriatal inputs in promoting avoidance and homeostatic feeding behavior.
- The ordering of the supplemental data makes it a little bit difficult to follow. I understand clumping all the inactive lever presses into one figure but other figures are out of order relative to how the data are presented in the main text and figures.

Reviewer #2 (Remarks to the Author):

Vollmer et al. report the results of an interesting series of experiments providing much needed new knowledge on the organization of the thalamostriatal pathway from paraventricular thalamus to nucleus accumbens in reward seeking behavior. The authors show, quite convincingly, that this circuit involves inputs onto parvalbumin neurons in the accumbens, is enriched in calcium permeable AMPA receptors, is opioid sensitive, and generally inhibits reward seeking across a variety of environmental conditions. These are all important knowledge gains.

There is a lot of work in this manuscript, some of it is very clever, all of it is very well done, and the

manuscript is very well presented. It significantly extends the field. I think the manuscript will be of interest to many readers and that it adds critical new knowledge. I enjoyed reading it and I think others will too. I had only the following comments on the manuscript:

1. It is not clear to me that the manuscript speaks to “risky reward motivated behaviors”. For example, the predator odor is presented in the home cages, not when mice are seeking sucrose. So, there is little to no ‘risk’ here. There is stress etc, but risk implies an adverse consequence of seeking sucrose. Moreover, the authors show that the same circuit mechanism suppress sucrose seeking after extinction training, which clearly has no risk at all. I am well persuaded by these data that this circuit is a general one for suppressing reward seeking behaviour (as predicted by others), but I am less persuaded that the reader learns much about risk. I suggest that “risky” is removed from the title, abstract etc.

2. “goal-directed’. In a number of places, the selective effects of manipulations on the ‘active’ lever and no effect on the ‘inactive’ lever are used to support the claim that behavior and the effects of manipulations are ‘goal directed’. This is hard to evaluate because there is no actual test of the goal directedness of the behavior here (e.g., contingency degradation; outcome devaluation). Moreover, it appears the sucrose delivery tube was physically located under the ‘active’ lever (Figure 1a), which reduces the utility and relevance of the ‘inactive’ lever to the task. I think what the authors mean is that manipulations were lever or behaviorally specific. I am certainly persuaded of this. It may be more helpful to use these terms rather than imputing untested mental states to the animals.

3. The decoder results are interesting, but these are based on only using the 1s before each lever press vs a random 1-s control. I wondered why only a 1s period was used and why this 1s was chosen over other possible durations? This could be better justified. Also, did the authors test a decoder using any other pre-lever press durations? If yes, how did they control for these exploratory analyses in their final statistical analyses?

4. The identification of these effects to PV interneurons is neat. PV neurons are sparse in the accumbens, so it is interesting that their manipulations here have such strong effects. However, the reader never learns how selective the hM4Di was to PV neurons in the accumbens in Figure 3i. Can the authors confirm that other cells in the accumbens did not express the DREADD?

Minor

1. I may have missed it, but do the authors state the number of cells in each cluster in Figure 1i or Figure 4? These are worth clearly stating.

2. State dose of DAMGO used in the manuscript proper.

3. Line 240: should refer to Figure 5H not 5F

Reviewer #3 (Remarks to the Author):

The manuscript by Vollmer et al. describes the role of PVT-NAc projection in the suppression of sucrose seeking. Using the head-fixed sucrose seeking paradigm, authors performed two-photon imaging and optogenetic manipulation of PVT neurons projecting to NAc. Furthermore, the authors examine the role of the opioid receptors in this neuronal type. Overall, all experiments are well designed and performed. However, some interpretation is not fully convincing.

1. It is difficult to connect the contribution of PVT projection to NAc PV neurons with behavioral changes in sucrose seeking. Especially, since several papers showed that PVT inputs preferentially modify D2-MSN in opioid-induced behavioral changes, it is not clear how the contribution of PVT projection is specifically modulating NAc PV neurons in sucrose seeking. Unless authors can image three different neurons (D1-MSN, D2-MSN, PV neurons) while modulating PVT inputs, this interpretation is not fully convincing. Moreover, since it is very likely that the chemogenetic inhibition of PV neurons (Figure 3k-o) will have massive effects on NAc circuitry already, the optogenetic stimulation of PVT together with PV chemogenetic manipulation may not be PV neuron dependent.

2. Authors found that CP-AMPA receptors are selectively located in PV neurons in NAc. However, the authors didn't show any effort to describe the potential roles of these CP-AMPA receptors on sucrose seeking. What is the role of PVT inputs for CP-AMPA receptor dependent PV neuronal activity?

3. The effects of DAMGO application on synaptic transmission (Figure 5) can be due to the postsynaptic effects since NAc MSNs also express MORs. More extensive analysis of synaptic transmission is needed to fully validate the presynaptic roles of MORs in PVT axon fibers. Again, the behavioral effects of DAMGO (Figure 5h) can be also due to the postsynaptic effects due to the DAMGO-induced changes in postsynaptic MSNs.

4. The systemic injection of heroin can induce neural adaptation in other brain areas, not just PVT-NAc inputs. These other adaptations can change the efficacy of PVT-NAc stimulation with heroin injection. Thus, it is difficult to know how the heroin injection can reverse the effect of PVT-NAc stimulation clearly. Authors should show how PVT-NAc projections are specifically changed by the systemic heroin injection.

5. The effect of the optogenetic manipulation of PVT to NAc projection on TMT or Yohimbe mediated suppression of sucrose seeking was not examined.

6. Authors can fully take advantage of two-photon imaging. The longitudinal imaging of single neurons can provide the trends in the adaptation of individual neurons.

Vollmer, Green et al., 2022
Nature Communications Resubmission
Response to Reviewers

We would like to thank the Reviewers for their outstanding suggestions, which we have addressed point-by-point below (specific concerns are in **BOLD** text, followed by our responses and additions to the manuscript in *ITALICIZED* font). In response to these concerns and suggestions, we provide new data, analyses, and clarifications which we feel have significantly strengthened the manuscript.

Reviewer #1 (Recommendations for the authors):

The present study by Vollmer, Green et al. test the hypothesis that a thalamostriatal circuit inhibits reward seeking during exposure to fear-provoking stimuli and that opioid receptor activation promotes risky behavior in the face of these stimuli by inhibiting this circuit via a presynaptic site of action. First the authors demonstrate that thalamostriatal PVT cells display heterogenous calcium responses to lever press activity (type 1 and 3 responses) associated with sucrose administration, and some ensembles are inhibited in a tonic manner. The authors demonstrate that optogenetic activation of PVT-NAcc pathway inhibited sucrose self-administration, while optogenetic inhibition of thalamostriatal neurons reversed TMT, yohimbine, and extinction-mediated reductions in sucrose self-administration. The authors subsequently demonstrated that thalamostriatal neurons innervate MSNs and PV interneurons, with the latter containing CP-AMPA receptors. The authors then show that the effects of optogenetic activation of PVT to NAcc afferents on sucrose self-administration is blocked by antagonism of CP-AMPA receptors in the NAcc and chemogenetic inhibition of NAcc PV neurons. The authors claim that thalamostriatal neurons and their terminals in the NAcc express MOR and that acute heroin administration decreases ensemble decoding of sucrose self-administration and blocks the suppressive effects of PVT to NAcc pathway stimulation and suppression induced by fear provoking stimuli. Lastly, the authors demonstrate that intra-NAcc DAMGO injections inhibit optogenetic -, TMT- and yohimbine-induced decreases in sucrose self-administration. They show that synapses from PVT to NAcc PV neurons are inhibited by DAMGO and that MOR deletion from PVT blocks the ability of intra-NAcc DAMGO from reversing the inhibitory effects of thalamostriatal optogenetic stimulation on sucrose self-administration. Overall, the study is of importance and tackles an important question. The authors test their hypothesis using an impressive set of interdisciplinary approaches. However, the authors make strong conclusions about their data that are not entirely supported by the present findings. Specifically, there are significant gaps that need to be addressed for the conclusions in their present form to be drawn, which include additional experiments and/or dialing back some of the interpretations and discussing caveats and alternative hypotheses. Below are some comments that will aid the authors in strengthening their manuscript and improving the cohesiveness of the study.

We want to thank Reviewer #1 for their excitement for our findings and very helpful feedback, which has aided us in improving the quality of our paper. Based on the concerns raised below, we have added 10 new experiments as well as new data analysis for existing imaging datasets. Furthermore, we have adjusted the language of the manuscript to ensure that caveats are addressed and more appropriately acknowledged.

1. There is a disconnect between the data across the different figures. The authors make strong claims that opioids acting through presynaptic receptors disrupt PVT inhibition, but this is largely decoupled from recordings of cell bodies and the effect of heroin presented earlier in the paper. Opioid receptors on terminals would inhibit terminal release

and any potential influence on cell body activity in thalamostriatal neurons would likely be an indirect consequence. That is, is there any relationship between what MORs are doing at terminals and what is happening in the cell bodies of thalamostriatal neurons? Moreover, it is unclear whether the exciting findings in Fig 5 are related to the findings with heroin. Additional studies linking the role of MORs on PVT terminals, activity of thalamostriatal cells, and how this is involved in heroin's effects would be needed to reach the interpretations presented, such as deleting MOR from thalamostriatal neurons and determining if that impacts the effect of heroin.

We agree with the Reviewer and have gone to great lengths to demonstrate that the findings across these figures (now Figure 4-6) are related. First, we use two-photon imaging and slice electrophysiology to show that both heroin injection and DAMGO application (experiments 1,2) reduces the activity/excitability of PVT→NAc somata and (experiment 3) reduces downstream synaptic inputs to NAc neurons in a manner that is prevented by Cre-dependent knockout of PVT μ -ORs (see **Figure 4, Figure 6, Supplementary Figure 9**). Second, we now show that behavioral disinhibition caused by (experiments 4-6) heroin injection or (experiments 7-8) intra-NAc DAMGO infusion is reversed by Cre-dependent μ -OR knockout in PVT, regardless of the behavioral suppressor administered (see **Figure 5; Figure 6**). These exciting new results lead us to conclude that μ -ORs on PVT→NAc neurons (possibly both somatic and on axon terminals) are required for opioid-driven behavioral disinhibition. However, we cannot rule out that opioids could also act elsewhere to drive behavioral disinhibition, including post-synaptically in NAc. Thus, we have lightened the language of the manuscript and include additional discussion (see page 8):

“Despite our findings that PVT μ -OR knockout prevents systemic heroin or intra-NAc DAMGO infusions from disinhibiting sucrose-seeking behaviors, caveats regarding the specificity of our results should be considered. First, we cannot dissociate whether opioid-driven μ -OR activation on PVT somata or PVT→NAc axon terminals are required for our observed behavioral effects. Considering that heroin and DAMGO can dramatically reduce activity at both PVT→NAc somata and downstream synapses, it is possible that both mechanisms are involved. Second, additional mechanisms could also contribute to opioid-induced behavioral disinhibition, such as μ -OR activation elsewhere in the brain and in other NAc cell types that express μ -ORs⁵⁴⁻⁵⁶. Overall, while our data suggests that opioid-induced inhibition of PVT→NAc neurons can disinhibit maladaptive behavioral actions, whether these effects are isolated to PVT→NAc somata and/or synapses has yet to be established.”

2. The authors should discuss the caveat that IEM-1640 may be acting on other cell types not examined in the present paper or that CP-AMPA receptors may become engaged downstream of PVT inputs to NAcc, including the PV neurons. The authors' interpretation should be much more cautious without additional evidence.

We agree. While our data showing that NAc CP-AMPA blockade results in behavioral disinhibition led us to examine PV interneuron involvement, we do not definitively state that CP-AMPA receptors on PV-interneurons are mediating the effects of IEM-1460. Furthermore, we now address caveats within the results (see page 5):

“While we therefore hypothesize that PVT→NAc neurons act selectively at CP-AMPA-enriched synapses on PV interneurons to suppress behavior, it should be noted that other synapses and NAc cell types may also be involved (see discussion).”

As well as the discussion (see page 7):

“While we used pharmacology and chemogenetics to target CP-AMPA receptors and PV interneurons, respectively, it is possible that these methods could have off-target effects. For example, non-PV cells within NAc could express CP-AMPA receptors, and thus CP-AMPA antagonism may act on other FSIs or non-PV cell types.”

3. Caveats associated with electrophysiological characterization of synaptic connections onto NAcc neurons should be discussed/addressed. A between subjects design comparing MSNs and PV neurons is subject to variability from ChR2 expression, and as such additional measures would be useful for determining synaptic strength differences between cell types. Moreover, since virus was used to label D2 cells the caveat that unlabeled cells may constitute putative D1 and unlabeled D2 cells should be addressed. That PV neurons display larger currents relative to MSNs is not surprising, since PV interneurons are broadly characterized by the presence of CP-AMPA receptors and enhanced excitatory synaptic strength in striatal structures and cortex. Furthermore, NAcc PV neurons have been implicated in mediating aversive behavior independent of the PVT (see work by Morales group (NIDA) for example).

We agree and have lightened the language of the manuscript to reflect putative targeting of D1-MSNs. Furthermore, we have added the following paragraph to the discussion which addresses the design of this electrophysiological experiment (see page 7):

“Our electrophysiological data show that accumbal PV interneurons, as compared to putative D1- and D2-MSNs, receive elevated excitatory drive from PVT neurons, although there are potential caveats to our viral targeting techniques. First, we used D2-Cre and PV-Cre transgenic mice to target MSNs or PV interneurons, respectively, which could have led to variability in ChrimsonR expression between groups of animals. Second, in our D2-Cre transgenic mice, we classified non-fluorescent neurons as putative D1-MSNs, whereas these cells could have been unlabeled D2-MSNs or other cell populations. Despite these caveats, our findings are consistent with previous literature showing that accumbal fast-spiking interneurons (FSIs) receive greater excitatory input from PVT as compared with undefined MSNs using a within-subject design³⁰. However, further studies comparing PVT synaptic input to each specific cell-type, including other subclasses of interneurons, within subjects could improve our understanding of PVT→NAc circuit biology.”

4. MOR activation may also inhibit glutamate release onto non-PV targets (e.g. MSNs), which may be contributing to the observed behavioral effects. The authors do not examine these synapses and make strong conclusions about PVT inputs to PV that warrant further

investigation. Furthermore, their physiology does not directly demonstrate presynaptic effects with the data presented, though it's implied.

This was an excellent point made by the Reviewer which we now address with new data and additional discussion. Using patch clamp electrophysiology, we confirm that DAMGO decreases PVT excitatory input to both accumbal PV-INs and MSNs in a manner that is blocked by Cre-dependent knockout of PVT μ -ORs (see new **Figure 6e-h**). Thus, μ -OR activation could be driving behavioral disinhibition through reduced PVT synaptic input to PV-INs, MSNs, or other cell types in NAc. However, our substantial data showing that PVT \rightarrow NAc^{PV-IN} circuitry is necessary for the suppression of sucrose seeking suggests that this pathway is likely involved. Nonetheless, we have significantly lightened the language of the manuscript to ensure that one pathway over another is not assumed, and we add additional discussion in this regard (see page 7, text revisions in response to concern #1 above).

5. In fig 1, The authors describe a tonic decrease in ensemble 2 and 3 activity over the course of the sucrose SA session that is blocked by fear provoking stimuli and extinction and reappears upon reinstatement. However, the emphasis is placed on ensemble 3 activity aligned to lever pressing based on decoding accuracy. Further discussion on this would be useful for the reader.

We would like to thank the Reviewer for this comment, and we have therefore adjusted some of the paragraph describing these imaging results to ensure that each ensemble is appropriately addressed without bias on ensemble 3.

6. The authors state that ensemble 1 and 3 decoding accuracy is decreased after heroin due to decreased changes in calcium activity, which is consistent with 4H. However, 4G suggest that decoding accuracy in ensemble 1 is increased relative to saline day. The same mismatch appears for ensemble 2, but not as pronounced as ensemble 1.

This is a fantastic catch by the Reviewer that was due to some data analysis imperfections, which we have since found and fixed. Specifically, for the CDF plots we were calculating the decoding values for each neuron as the unshuffled decoding score minus the averaged shuffled decoding score across all neurons, such that chance decoding would equal 0. This is an issue as different neurons and clusters are going to have different shuffled values that deviate from one another due to the number of lever presses within a session (specifically, low presses generally result in higher shuffled decoding scores). The summarized heatmap therefore deviated from the results in the CDF plots as each cluster's decoding score was analyzed as its own unshuffled vs shuffled score within the heatmap (using a simple t-test). We therefore have addressed the above data analysis concerns in a few ways:

- 1) We have taken out the heatmap, which served as a poor descriptor of the data.
- 2) We use shuffled decoding values for each neuron for decoding score normalization.
- 3) We use single-cell tracking to analyze the same neurons between the two behavioral sessions, such that between-cell and between-cluster variability is accounted for (this is also in response to Reviewer #3, Concern #6).
- 4) We show that both the activated and inhibited ensembles display significant response adaptation following heroin injection (see **Figure 4h**).

- 5) We show both population and cluster-specific decoding within CDF plots (see new **Figure 4i, j**), revealing that decoding scores are significantly decreased at the population level and for cluster #3 (see new **Supplementary Fig. 7c-e**).

7. The switch to real time place preference evoked by opto PVT to NAcc stim with heroin injection is very interesting and in some ways may be relevant to the potentiation of sucrose self-administration that is observed in response to opto stim, TMT, and yohimbine in fig 4 and 5. This suggest that MOR activity may be unmasking a population of pro-sucrose administration neurons upon inhibition of MOR sensitive neurons.

We agree with the Reviewer about this interesting finding, and have added additional language in the results to highlight its importance (see page 5):

“Together, these data suggest that systemic opioids may be modulating PVT→NAc neuronal activity, such that sucrose self-administration is promoted, rather than inhibited, in the presence of behavioral suppressors.”

8. In supplementary figure s2, cluster 1 seems to contain a subset of cells that are briefly inhibited prior to lever press and are excited during reward delivery late in acquisition and inhibited during reinstatement, suggesting they are distinct from other cells in cluster 1. This cluster in some ways has both cluster 1 and 3 properties.

This was an excellent observation by the Reviewer. We have added and adjusted language in the discussion to include this point (see page 8):

“...though we cluster PVT→NAc neurons by activity (see Fig. 1), each ensemble seems to contain subsets of cells with distinct responses, despite isolating PVT neurons by location (i.e., posterior) and connection (i.e., projections to NAc).”

9. The MOR staining of PVT neurons and their terminals could be due to chance alone as MOR-like immunoreactivity appears to be widespread across the image. In the zoom within the representative images it appears that background from adjacent regions was cropped or subtracted. Immunostaining for MOR with this in MOR loxP mice would be useful for validating the antibody and genetic approach.

This is a fantastic point that we have addressed through several experiments and adjustments to the figures/text.

First, we show new example IHC that we used to validate Cre-dependent μ -OR knockout in μ -OR loxP mice (**New Supplementary Figure 8**). This IHC experiment was performed using a different antibody, which was originally used by others to validate μ -OR knockout in μ -OR loxP mice (Cui *et al.* 2014 PMID: 24413699). We therefore replaced the original images of PVT somata with those taken using the new antibody, and we do not perform the same background subtraction based on the Reviewer’s excellent point. Furthermore, we do not include images of PVT→NAc axon colocalized with μ -OR IHC as overlap could be due to chance.

Second, we show example images and quantification from a new RNAscope experiment that was used to demonstrate Cre-dependent knockout of μ -OR mRNA (see new **Figure 5a-c**).

Third, we show that PVT neuronal excitability and synaptic input to downstream NAc neurons are

Vollmer, Green et al., 2022
Nature Communications Resubmission
Response to Reviewers

inhibited by the μ -OR agonist DAMGO, effects that are blocked by Cre-dependent knockout of PVT μ -ORs (new **Figure 6e-h**; **Supplementary Figure 9**).

Altogether, these experiments justify the use of the antibody, μ -OR loxP mice, and the point that μ -ORs are expressed and functional within PVT \rightarrow NAc projection neurons.

10. NAcc PV cells are sparse relative to other interneuron populations. Moreover, PV-Cre mice used in the present study have incomplete genetic penetrance in the NAcc and may select for subtypes of NAcc fast-spiking interneurons in addition to non-PV cells with CP-AMPA receptors (see Adam Carter's work and Yan Dong). The authors should mention the incomplete penetrance to let the reader know that this may pertain a selective subpopulation of PV cells and overall broaden the discussion of how PV cells are limited in density in NAcc relative to striatum in the context of the present work.

We agree and have now added a paragraph to the discussion section to address these concerns (see page 7-8):

“Our data support the idea that accumbal PV interneurons and CP-AMPA receptors are necessary for the suppression of sucrose self-administration. Previously, others have shown that accumbal PV interneurons, as well as other FSIs within NAc, can act as powerful regulators of local neuronal activity and behavior despite being sparsely distributed^{41–45}. While we used pharmacology and chemogenetics to target CP-AMPA receptors and PV interneurons, respectively, it is possible that these methods could have off-target effects. For example, non-PV cells within NAc could express CP-AMPA receptors, and thus CP-AMPA receptor antagonism may act on other FSIs or non-PV cell types. Furthermore, it is possible that our targeting of PV interneurons could have profound effects on downstream neurophysiology, and therefore may not be completely selective for our circuit-of-interest. Finally, our PV interneuron cell targeting is likely to select for only a subpopulation of PV-expressing neurons due to incomplete genetic penetrance of the PV-Cre transgenic mouse line^{46,47}. Notably, our electrophysiological recordings suggest that our Cre-dependent targeting of PV interneurons at least selects for FSIs, as fluorescent cells within PV-Cre mice displayed fast-spiking properties. Additionally, we find that these cells are inwardly rectifying, suggesting the presence of CP-AMPA receptors. Despite these findings, future studies selectively targeting CP-AMPA receptors at PVT \rightarrow NAc^{PV-IN} synapses could elucidate the precise role of these receptors for PVT \rightarrow NAc-dependent behavioral suppression.”

11. The authors should show the lever press – aligned population activity during the fear-provoking stimuli in addition to the change across the session that is reported.

While we agree with the Reviewer that this would be interesting to show, we have not included this in our revision. First, our *in vivo* imaging heatmaps are generated by averaging neuronal activity across active lever presses and subjects. Neuronal activity is time-locked to the active lever press and, due to the low number of active lever presses observed on days where mice were exposed to behavioral suppressors (e.g., TMT, yohimbine, extinction learning), it is difficult not possible to generate a heatmap that accurately depicts lever pressing during behavioral

Vollmer, Green et al., 2022
Nature Communications Resubmission
Response to Reviewers

suppression. Thus, we decided the most accurate way to present changes in PVT→NAc activity during behavioral suppression was by showing the change in fluorescence across sessions.

12. The intrinsic properties of the MSNs in the representative traces do not show the well-characterized regular spiking properties of healthy MSNs ex-vivo, raising the possibility the cells / preparation was not optimal.

We agree and would like to reassure the Reviewer that *ex vivo* cell preparation was fine. The lead PI has performed slice recordings for many years (including in striatum) and identical methodologies were used in this case. The issue is likely that the previous waveforms were from sweeps wherein a very large current pulse was administered (>250pA; in attempt to show the very high frequency of firing in PV interneurons vs spike adaptation in MSNs). Thus, we have now adjusted the waveforms to show spiking in each cell type in response to a moderate current injection (50pA; see **New Figure 3B**).

13. The authors should cite relevant work on thalamostriatal circuitry where appropriate from the Penzo group at NIMH describing a role for thalamostriatal inputs in promoting avoidance and homeostatic feeding behavior.

Although we had cited Penzo et al., 2015, Beas et al. 2018, and Gao et al., 2020 (each from the Penzo group), we have now added Beas et al., 2020.

14. The ordering of the supplemental data makes it a little bit difficult to follow. I understand clumping all the inactive lever presses into one figure, but other figures are out of order relative to how the data are presented in the main text and figures.

We apologize for any confusion surrounding the Supplementary Figures. We have now distributed the inactive lever pressing graphs across Supplementary Figures, such that it aligns with how the data are presented within the main text.

Reviewer #2 (Recommendations for the authors):

Vollmer et al. report the results of an interesting series of experiments providing much needed new knowledge on the organization of the thalamostriatal pathway from paraventricular thalamus to nucleus accumbens in reward seeking behavior. The authors show, quite convincingly, that this circuit involves inputs onto parvalbumin neurons in the accumbens, is enriched in calcium permeable AMPA receptors, is opioid sensitive, and generally inhibits reward seeking across a variety of environmental conditions. These are all important knowledge gains.

There is a lot of work in this manuscript, some of it is very clever, all of it is very well done, and the manuscript is very well presented. It significantly extends the field. I think the manuscript will be of interest to many readers and that it adds critical new knowledge. I enjoyed reading it and I think others will too. I had only the following comments on the manuscript:

We would like to thank Reviewer #2 for their excellent suggestions and excitement for our study. We have responded to each concern below, through the addition of new experiments, analyses, discussion points, and consideration of alternative interpretations.

1. It is not clear to me that the manuscript speaks to “risky reward motivated behaviors”. For example, the predator odor is presented in the home cages, not when mice are seeking sucrose. So, there is little to no ‘risk’ here. There is stress etc, but risk implies an adverse consequence of seeking sucrose. Moreover, the authors show that the same circuit mechanism suppress sucrose seeing after extinction training, which clearly has no risk at all. I am well persuaded by these data that this circuit is a general one for suppressing reward seeking behavior (as predicted by others), but I am less persuaded that the reader learns much about risk. I suggest that “risky” is removed from the title, abstract etc.

We agree and have changed the title according to the Reviewer’s recommendation:

“An opioid-gated thalamoaccumbal circuit for the suppression of reward seeking”.

Additionally, we have removed language from the manuscript wherein “risky” was used inappropriately.

2. “Goal-directed”. In a number of places, the selective effects of manipulations on the ‘active’ lever and no effect on the ‘inactive’ lever are used to support the claim that behavior and the effects of manipulations are ‘goal directed’. This is hard to evaluate because there is no actual test of the goal directedness of the behavior here (e.g., contingency degradation; outcome devaluation). Moreover, it appears the sucrose delivery tube was physically located under the ‘active’ lever (Figure 1a), which reduces the utility and relevance of the ‘inactive’ lever to the task. I think what the authors mean is that manipulations were lever or behaviorally specific. I am certainly persuaded of this. It may be more helpful to use these terms rather than imputing untested mental states to the animals.

We thank the Reviewer for bringing up these issues. We have changed the language within the manuscript to describe our manipulations as being lever-specific, rather than goal directed.

Vollmer, Green et al., 2022
Nature Communications Resubmission
Response to Reviewers

Furthermore, we apologize for any ambiguity within previous versions of Figure 1a. We have edited the figures so that the sucrose delivery tube appears to be directly in-between the active and inactive levers. We have also added language to clarify the sucrose spout position within the methods:

Under Head-fixed behavior, we report (page 24):

“Sucrose delivery spouts were arranged equally between both levers so that mice would not be biased toward either lever.”

3. The decoder results are interesting, but these are based on only using the 1s before each lever press vs a random 1-s control. I wondered why only a 1s period was used and why this 1s was chosen over other possible durations? This could be better justified. Also, did the authors test a decoder using any other pre-lever press durations? If yes, how did they control for these exploratory analyses in their final statistical analyses?

This is an excellent question that deserves justification within the text, which we now add (see page 26):

“We used a 1-second epoch prior to the lever press based on: (1) a pre-lever press epoch would ensure that the decoding was not due to cue presentation, liquid delivery, or sucrose consumption and (2) past studies have demonstrated single-cell calcium events during a 1-second pre-reward trace interval can be used to accurately predict reward learning within a Pavlovian conditioning task^{6,26}. Thus, a 1-second epoch immediately before lever pressing seemed most appropriate for our decoding analysis and was chosen beforehand such that only one analysis was performed.”

4. The identification of these effects to PV interneurons is neat. PV neurons are sparse in the accumbens, so it is interesting that their manipulations here have such strong effects. However, the reader never learns how selective the hM4Di was to PV neurons in the accumbens in Figure 3i. Can the authors confirm that other cells in the accumbens did not express the DREADD?

We agree and have now added new data validating the use of the DIO-hM4Di-DREADD-mCherry for CNO-induced inhibition of NAc PV interneurons (see new **Supplemental Figure 6**). We specifically find that mCherry-expressing neurons display electrophysiological properties consistent with PV interneurons, and these cells are inhibited by bath application of CNO. In contrast, neighboring mCherry-negative neurons were not inhibited by CNO and show electrophysiological properties consistent with other NAc cells (in particular, medium spiny neurons).

Minor:

1. I may have missed it, but do the authors state the number of cells in each cluster in Figure 1i or in Figure 4? These are worth clearly stating.

We agree and have now stated the number of cells in each cluster within the captions for Figures 1 and 4.

Vollmer, Green et al., 2022
Nature Communications Resubmission
Response to Reviewers

2. State dose of DAMGO used in the manuscript proper.

We agree and have stated the dose of DAMGO used for intracranial infusions within the manuscript proper.

3. Line 240: should refer to Figure 5H not 5F.

We thank the Reviewer for pointing this out and have corrected this mix-up.

Reviewer #3 (Recommendations for the authors):

The manuscript by Vollmer et al. describes the role of PVT-NAc projection in the suppression of sucrose seeking. Using the head-fixed sucrose seeking paradigm, authors performed two-photon imaging and optogenetic manipulation of PVT neurons projecting to NAc. Furthermore, the authors examine the role of the opioid receptors in this neuronal type. Overall, all experiments are well designed and performed. However, some interpretation is not fully convincing.

We would like to sincerely thank Reviewer #3 for their enthusiasm for the paper as well as their concerns about some of our data interpretation. In response, we both added experiments and dialed back some of the interpretation which we agree was over-extended.

1. It is difficult to connect the contribution of PVT projection to NAc PV neurons with behavioral changes in sucrose seeking. Especially, since several papers showed that PVT inputs preferentially modify D2-MSN in opioid-induced behavioral changes, it is not clear how the contribution of PVT projection is specifically modulating NAc PV neurons in sucrose seeking. Unless authors can image three different neurons (D1-MSN, D2-MSN, PV neurons) while modulating PVT inputs, this interpretation is not fully convincing. Moreover, since it is very likely that the chemogenetic inhibition of PV neurons (Figure 3k-o) will have massive effects on NAc circuitry already, the optogenetic stimulation of PVT together with PV chemogenetic manipulation may not be PV neuron dependent.

We would like to thank the Reviewer for this point, which we agree with overall and now address through easing of data interpretation and additional discussion. Specifically, while we predict that PVT→NAc stimulation causes behavioral disinhibition through the activation of downstream PV-INs (considering the experimental evidence within our paper and supporting studies), we provide several critical caveats. Furthermore, while we conclude that opioids can act at μ -opioid receptors at PVT→NAc axons, somata, or both to drive behavioral disinhibition, we note that this could be due to a reduction in activity at a variety of PVT→NAc synapses and downstream neurons, and not necessarily at PVT→NAc^{PV-IN} synapses specifically.

We do feel it is important to consider several other points. First, we show that intra-NAc infusion of a CP-AMPA antagonist, but not D1 or D2 receptor antagonist, abolishes the suppression of sucrose seeking caused by PVT→NAc stimulation. Considering the specificity of these receptors for FSIs/PV interneurons (Gittis et al., 2011 PMID: 22049415; Manz et al., 2020 PMID: 32726634) and PVT→NAc^{PV-IN} but not PVT→NAc^{MSN} synapses (Figure 3), it is likely that the PVT→NAc^{PV-IN} pathway is important for our behavioral findings. This interpretation is even further supported, if not confirmed, by our chemogenetics data showing that inhibition of PV-interneurons reverses the behavior-suppressing effects caused by PVT→NAc stimulation, TMT exposure, and yohimbine exposure (Figure 3). It should be noted that these chemogenetic manipulations are less potent than optogenetic inhibition, as we see a modest and cell-type specific reduction in PV-IN excitability following CNO exposure (see new data in **Supplementary Fig. 6c**). Finally, as mentioned by Reviewer 1, PV interneurons represent a very sparse population of NAc neurons (~1-2% of cells), and thus we feel that behavioral effects found via PV interneuron inhibition are quite fascinating. It should be noted that D1-MSNs and D2-MSNs, which also have robust collaterals and projections for local and distal inhibition, are far more common than PV-INs (~95% of NAc neurons) and are often targeted via optogenetics and chemogenetics to identify their function for behavioral control.

Vollmer, Green et al., 2022
Nature Communications Resubmission
Response to Reviewers

Despite these points, we fully agree that there may be off-target effects caused by our pharmacological or chemogenetic manipulations. Thus, we have added additional discussion to highlight the Reviewer's concerns (for example, see page 7-8, text revisions in response to Reviewer #1, Concern #10 above):

“Our data support the idea that accumbal PV interneurons and CP-AMPArs are necessary for the suppression of sucrose self-administration. Previously, others have shown that accumbal PV interneurons, as well as other FSIs within NAc, can act as powerful regulators of local neuronal activity and behavior despite being sparsely distributed⁴¹⁻⁴⁵. While we used pharmacology and chemogenetics to target CP-AMPArs and PV interneurons, respectively, it is possible that these methods could have off-target effects. For example, non-PV cells within NAc could express CP-AMPArs, and thus CP-AMPAr antagonism may act on other FSIs or non-PV cell types. Furthermore, it is possible that our targeting of PV interneurons could have profound effects on downstream neurophysiology, and therefore may not be completely selective for our circuit-of-interest.”

Next, while we agree with the Reviewer that being able to modulate PVT while monitoring D1-MSNs, D2-MSNs, and PV interneurons would aid in elucidating the precise mechanism in which PVT→NAc neurons influence behavior, recording from 3 different cell types simultaneously during PVT stimulation is simply not feasible with current technologies available. As such, it is not possible for us to complete this experiment.

2. Authors found that CP-AMPArs are selectively located in PV neurons in NAc. However, the authors didn't show any effort to describe the potential roles of these CP-AMPArs on sucrose seeking. What is the role of PVT inputs for CP-AMPArs dependent PV neuronal activity?

We used a combination of optogenetics and neuropharmacology to show that CP-AMPArs are required for PVT→NAc dependent suppression of sucrose seeking (Figure 3j). While CP-AMPArs are not the focus of the manuscript, these experiments provided initial support for the idea that PVT→NAc PV interneuron synapses may be involved in the suppression of sucrose seeking. This idea was then strongly supported by the subsequent chemogenetics experiments, wherein we combine optogenetic manipulation of PVT→NAc neurons with chemogenetic manipulation of PV interneurons. We have also added additional discussion highlighting the previous findings that PV interneurons within NAc can act as behavioral regulators (see page see page 7-8, text revisions in response to Reviewer #1, Concern #10 above).

3. The effects of DAMGO application on synaptic transmission (Figure 5) can be due to the postsynaptic effects since NAc MSNs also express MORs. More extensive analysis of synaptic transmission is needed to fully validate the presynaptic roles of MORs in PVT axon fibers. Again, the behavioral effects of DAMGO (Figure 5h) can be also due to the postsynaptic effects due to the DAMGO-induced changes in postsynaptic MSNs.

This is an excellent point, which we now add experiments and discussion to address:

- a) Synaptic physiology: We have added **new synaptic physiology experiments**. Using Cre-dependent knockout of μ -ORs in PVT, we find that DAMGO no longer reduces

PVT→NAc synaptic transmission (new **Figure 6h**). These data suggest that the effect of DAMGO on optically evoked EPSCs at PVT→NAc synapses requires presynaptic μ -ORs.

- b) **Behavioral findings:** We have added **5 additional behavioral experiments** to highlight the requirement for PVT μ -ORs for opioid-induced behavioral disinhibition. While we confirm that heroin disinhibits sucrose seeking regardless of the behavioral suppressor presented (i.e., PVT→NAc stimulation, TMT, or yohimbine), we show that knockout of PVT μ -ORs prevents heroin-induced behavioral disinhibition (new **Figure 5d-f**). Furthermore, we show that PVT μ -OR knockout prevents NAc DAMGO infusions from causing TMT- and yohimbine-induced behavioral disinhibition (previously we had only examined PVT→NAc dependent behavioral disinhibition; new **Figure 6k, l**). Overall, these findings confirm that the effects of systemic heroin and intra-NAc DAMGO on behavioral disinhibition require PVT μ -ORs.
- c) Discussion: As rightly pointed out by the Reviewer, our experiments simply cannot rule out the possibility that DAMGO also has postsynaptic effects on PVT→NAc neurophysiology and thus behavior. Thus, while we indicate that PVT μ -ORs are indeed required for our observed findings, we dial back the interpretation throughout the manuscript and add discussion points to this effect (e.g., see page 8):

“...additional mechanisms could also contribute to opioid-induced behavioral disinhibition, such as μ -OR activation elsewhere in the brain and in other NAc cell types that express μ -ORs⁵⁴⁻⁵⁶. Overall, while our data suggests that opioid-induced inhibition of PVT→NAc neurons can disinhibit maladaptive behavioral actions, whether these effects are isolated to PVT→NAc somata and/or synapses has yet to be established.”

4. The systemic injection of heroin can induce neural adaptation in other brain areas, not just PVT-NAc inputs. These other adaptations can change the efficacy of PVT-NAc stimulation with heroin injection. Thus, it is difficult to know how the heroin injection can reverse the effect of PVT-NAc stimulation clearly. Authors should show how PVT-NAc projections are specifically changed by the systemic heroin injection.

This is another outstanding point that we now address with several new experiments and discussion:

- a) Using two-photon calcium imaging we demonstrate that PVT→NAc neurons decrease activity and encoding of sucrose seeking following systemic heroin injection (Figure 4a-j).
- b) Using patch-clamp electrophysiology we add new data showing that μ -OR activation diminishes evoked activity in PVT→NAc neurons, an effect that is prevented by the knockout of PVT μ -ORs (new **Supplementary Figure 9**).
- c) It remains true that the behavioral effects caused by heroin could be due to adaptations on or upstream of PVT→NAc neurons. Thus, we knocked μ -ORs out of PVT, and show that heroin-induced behavioral disinhibition completely absent (see **new Figure 5**). Thus, the behavioral and physiological effects of heroin described in our study require PVT μ -ORs.

- d) Despite these findings, it remains possible that heroin could also be acting elsewhere to drive behavioral disinhibition and adaptations in neurophysiology, which we address through additions to the text such as that referenced above.

5. The effect of the optogenetic manipulation of PVT to NAc projection on TMT or Yohimbe mediated suppression of sucrose seeking was not examined.

In Figure 2, we bidirectionally manipulate PVT→NAc neurons following TMT- (Figure 2f) and yohimbine- (Figure 2g) mediated suppression of sucrose seeking.

6. Authors can fully take advantage of two-photon imaging. The longitudinal imaging of single neurons can provide the trends in the adaptation of individual neurons.

This is an excellent point, and we have now added new data tracking PVT→NAc neuronal activity across time. First, we track PVT→NAc responses across early to late sucrose self-administration. We find that PVT→NAc ensemble dynamics develop over learning, as ensembles 1 (activated) and 3 (inhibited) show a significant response adaptation over time (see new **Supplementary Figure 2c-f**). Next, we track PVT→NAc neuronal activity from saline to heroin injection tests, and find that PVT→NAc ensembles 1 and 3 have significant response attenuation during the heroin test such that the activity of these neurons can no longer be used to decode active lever pressing (see new **Figure 4f-j**; **Supplementary Fig 7c-e**).

REVIEWERS' COMMENTS

Reviewer #1 (Remarks to the Author):

The authors have adequately addressed my comments and concerns. Congratulations on the elegant and impactful study.

Reviewer #2 (Remarks to the Author):

The authors have made considerable revisions in response to my initial review. I thank them for this. I remain of the view that this is an interesting and important contribution to the literature on PVT function. I am less certain of the ultimate utility of simple lever and extinction type approaches for understanding core problems in reward seeking - and the field has largely moved on from these- but the kinds of insights generated by this manuscript are important and could lead to a revision of this view.

Reviewer #3 (Remarks to the Author):

The authors have been very responsive to my comments. They have added results from a number of experiments and also modified language to account for limitations.

I also feel that the authors have done a very thorough and careful job addressing concerns of other reviewers'.